# Index2Sort: Sorting Algorithm Using Static Index Data Structure

## Abstract

We introduce Index2Sort, a general framework for deriving sorting algorithms from static indexes. Index2Sort treats the index as an opaque box that exposes only two operations: index construction and rank queries. This abstraction allows Index2Sort to be applied to various index structures, including classical and learned indexes. Our theoretical analysis shows that the computational guarantees of the index transfer directly to Index2Sort. If the index can be constructed in expected time $\mathcal{O}(nC(n))$ and can answer rank queries in expected time $\mathcal{O}(Q(n))$, then Index2Sort sorts the input in expected time $\mathcal{O}(nC(n) + nQ(n))$. In particular, when using a state-of-the-art learned index with $C(n) = Q(n) = 1$, this yields an expected complexity of $\mathcal{O}(n)$, which is a strictly tighter bound than those of existing learned sorting algorithms. In contrast to recent theoretical works on learned sorting, which derive complexity guarantees by analyzing the internal structure of a learned index and designing a sorting algorithm with a similar structure, Index2Sort achieves stronger guarantees without requiring any inspection or modification of the index internals.

## 1    Introduction

Recent research integrating machine learning into classical data structures and algorithms has led to dramatic performance improvements in fundamental computational tasks, including indexing and sorting. This line of work has given rise to a new class of algorithms known as *learned indexes* (Kraska et al., 2018) and *learned sorts* (Kraska et al., 2019). Both share a common design principle: they approximate the cumulative distribution function (CDF) of the data with a machine learning model and leverage its predictions within the algorithm. In these areas, researchers have not only demonstrated significant empirical speedups but also developed algorithms with strong expected-time complexity guarantees under distributional assumptions (Zeighami & Shahabi, 2023; Croquevielle et al., 2025; Sato & Matsui, 2024; Zeighami & Shahabi, 2024).

There is a noticeable gap between the theoretical progress on learned indexes and that on learned sorts. Advances in learned sort have historically followed those in learned index, but with some delay. For example, after the development of static learned indexes with expected construction time $\mathcal{O}(n \log \log n)$ and expected rank query time $\mathcal{O}(\log \log n)$ (Zeighami & Shahabi, 2023), researchers carefully examined the internal structure of such learned indexes and redesigned them for sorting, eventually producing learned sorts with expected $\mathcal{O}(n \log \log n)$ time (Sato & Matsui, 2024; Zeighami & Shahabi, 2024). Later, theory on static learned indexes advanced further, achieving expected construction time $\mathcal{O}(n)$ and expected rank query time $\mathcal{O}(1)$ under similar assumptions (Croquevielle et al., 2025). However, this breakthrough has not yet translated to sorting; the best existing learned sorts remain bound by expected $\mathcal{O}(n \log \log n)$ time.

Bridging this gap is non-trivial because static indexing fundamentally depends on sorting. Static indexes require a sorted array as input for construction; thus, attempting to leverage the algorithms and theoretical guarantees of static indexes for sorting inevitably creates a circular dependency. In contrast, the connection between dynamic indexing and sorting is straightforward, as inserting elements into a dynamic index and performing an in-order traversal yields a sorted sequence. However, dynamic indexes are significantly more complex to design and historically lag behind their static counterparts. For instance, ALEX (Ding et al., 2020), a dynamic extension of the learned

index framework, was developed more than a year after the initial proposal of static learned indexes (Kraska et al., 2018).

This observation naturally raises two questions: (1) Can we overcome the inverse dependency of static indexes to design a learned sort with expected $\mathcal{O}(n)$ time under the same assumptions as (Croquevielle et al., 2025)? (2) If even stronger static learned indexes are developed in the future, can their improvements be automatically transferred to learned sorts?

Our answer to both questions is yes. In this work, we present **Index2Sort**, the first general framework that derives sorting algorithms from any static index. Index2Sort automatically inherits the computational guarantees of the underlying static index, thereby bridging the theoretical gap between learned static indexes and learned sorts. Specifically, if the static index can be constructed in expected time $\mathcal{O}(nC(n))$ and answer rank queries in expected time $\mathcal{O}(Q(n))$, then Index2Sort sorts the input in expected time $\mathcal{O}(nC(n) + nQ(n))$. As a concrete example, applying the state-of-the-art learned static index of Croquevielle et al. (2025) immediately yields a sorting algorithm with expected running time $\mathcal{O}(n)$ under standard distributional assumptions. Furthermore, thanks to the generality of Index2Sort, if learned static indexes with even stronger theoretical guarantees are developed in the future, their benefits will carry over directly to sorting.

Our contributions are summarized as follows:

- **General opaque-box sorting framework**: We propose Index2Sort, the first framework that performs sorting by treating any static index as an opaque box. This achieves a conceptual inversion of the usual dependency between static indexes and sorting: although a staic index is constructed over a sorted array, we demonstrate that it can itself be used for sorting.

- **Automatic inheritance of guarantees**: We formally prove that Index2Sort automatically inherits the computational guarantees of the underlying static index, thereby establishing a formal and general theoretical bridge between indexing and sorting.

- **State-of-the-art theoretical guarantees for sorting**: By instantiating Index2Sort with state-of-the-art learned indexes, we immediately obtain algorithms that achieve expected running time $\mathcal{O}(n)$ under the standard distributional assumptions, and we further show that $\mathcal{O}(n \log \log n)$ can still be achieved under even weaker assumptions. These results are provably stronger complexity guarantees than all existing learned sorting algorithms.

- **Future-proof paradigm**: Beyond these results, Index2Sort offers a paradigm that continuously benefits from progress in index research: any theoretical advance in indexing immediately translates into an advance in sorting.

This paper is organized as follows. Section 2 introduces the necessary definitions and notation, and Section 3 presents Index2Sort along with its complexity guarantees. Section 4 provides an experimental validation of these guarantees. Section 5 discusses related work, Section 6 examines limitations, and Section 7 concludes the paper.

## 2 PRELIMINARIES

Here, we introduce several definitions and notations required for our problem setup.

**Sorting.** Sorting is the operation that converts an input array into a sorted array. Let $\boldsymbol{x} = [x_1, x_2, \ldots, x_n]$ be an array of $n$ real numbers. The array $\boldsymbol{x}$ may contain duplicate elements; in other words, there may exist indices $i, j$ such that $x_i = x_j$. Sorting transforms $\boldsymbol{x}$ into $\boldsymbol{x}' = [x_{\pi(1)}, x_{\pi(2)}, \ldots, x_{\pi(n)}]$, where $\pi$ is a bijective function from $\{1, 2, \ldots, n\}$ to $\{1, 2, \ldots, n\}$ that satisfies $x_{\pi(i)} \leq x_{\pi(j)}$ for all $i, j$ such that $i < j$. In this paper, we use a prime symbol ($'$) on a vector (e.g., $\boldsymbol{x}'$) to denote its sorted version.

**Indexing.** Algorithms for static index data structures consist of two phases: the construction phase and the query response phase. In the construction phase, a sorted array $\boldsymbol{x}' \in \mathbb{R}^n$ is provided, and an index data structure is constructed. The constructed index does not necessarily support the insertion or deletion of elements. In the query response phase, the index data structure processes a given

query $q \in \mathbb{R}$ and returns the rank of $q$, that is, the number of elements in the array $\boldsymbol{x}'$ that are less than $q$. We assume only the above two functionalities of the index and make no other assumptions, such as the internal structure.

**Data Distribution and Distribution Shift.** The theoretical guarantees for learned indexes and learned sorts often rely on distributional assumptions. We introduce a notation for distributions and their distance metrics. We adopt definitions nearly identical to those in (Zeighami & Shahabi, 2024), which provide a consistent framework for describing assumptions about distributions.

We define an array $\boldsymbol{D} = [D_1, D_2, \ldots, D_n]$ as being *sampled independently from distributions* $\boldsymbol{\chi} = [\chi_1, \chi_2, \ldots, \chi_n]$ if each $D_i$ is drawn independently from $\chi_i$ for all $i = 1, 2, \ldots, n$. For brevity, we write this as $\boldsymbol{D} \sim \boldsymbol{\chi}$. When all elements of $\boldsymbol{D}$ are sampled i.i.d. from a single distribution $\chi$, we denote this as $\boldsymbol{D} \overset{\text{iid}}{\sim} \chi$. If $\boldsymbol{D} \sim \boldsymbol{\chi}$ and $\chi_i \in \mathfrak{X}$ for all $i$, we state that $\boldsymbol{D}$ is *sampled from the distribution class* $\mathfrak{X}$, where $\mathfrak{X}$ is a set of distributions. We define the following representative distribution classes:

- $\mathfrak{X}_{\rho_1, \rho_2}$ ($\rho_1 > 0, \rho_2 < \infty$): The set of distributions with probability density functions $f$ over a finite continuous domain $\mathcal{K}$ such that $\forall x \in \mathcal{K}, \rho_1 \leq f(x) \leq \rho_2$.

- $\mathfrak{X}_{\rho_f}$ ($\rho_f < \infty$): The set of distributions with probability density functions $f$ over a continuous finite domain $\mathcal{K}$ such that $\int_{\mathcal{K}} f^2(x)dx \leq \rho_f$.

- $\mathfrak{X}_C$ ($C > 0$): The set of subexponential distributions with the tail decay parameter $C$. Formally, if $X \sim \chi$ for some $\chi \in \mathfrak{X}_C$, then $\Pr[|X| \geq x] \leq 2e^{-Cx}$ for all $x \geq 0$.

The class $\mathfrak{X}_{\rho_1, \rho_2}$ appears in (Zeighami & Shahabi, 2023; Sato & Matsui, 2024; Zeighami & Shahabi, 2024), while $\mathfrak{X}_{\rho_f}$ and $\mathfrak{X}_C$ are used in (Croquevielle et al., 2025).

These classes form a hierarchy: bounded density implies a bounded $L_2$ norm, and a bounded $L_2$ norm on a finite domain implies sub-exponential tails.

**Fact 2.1.** *For any $\rho_1 > 0$ and $\rho_2 < \infty$, there exists $\rho_f < \infty$ such that $\mathfrak{X}_{\rho_1, \rho_2} \subseteq \mathfrak{X}_{\rho_f}$. Furthermore, for any $\rho_f < \infty$, there exists $C > 0$ such that $\mathfrak{X}_{\rho_f} \subseteq \mathfrak{X}_C$.*

*Proof.* First, we show $\mathfrak{X}_{\rho_1, \rho_2} \subseteq \mathfrak{X}_{\rho_f}$. Let $\chi \in \mathfrak{X}_{\rho_1, \rho_2}$ with probability density function $f$. Then, we have $\int_{\mathcal{K}} f^2(x)dx \leq \int_{\mathcal{K}} \rho_2 f(x)dx = \rho_2$. Thus, choosing $\rho_f = \rho_2$ satisfies the required condition.

Next, we show $\mathfrak{X}_{\rho_f} \subseteq \mathfrak{X}_C$. Let $\chi \in \mathfrak{X}_{\rho_f}$. By definition, $\chi$ is supported on a finite domain $\mathcal{K}$. Let $M = \sup_{z \in \mathcal{K}} |z| < \infty$. Setting $C = (\ln 2)/M$ yields $\Pr[|X| \geq x] \leq 2e^{-Cx}$ for all $x \geq 0$. This is because for $x \leq M$, we have $2e^{-Cx} \geq 1$, and for $x > M$, $\Pr[|X| \geq x] = 0$. $\qquad\square$

To quantify distribution shift, we use total variation distance as in (Zeighami & Shahabi, 2024). For a sequence of distributions $\boldsymbol{\chi} = [\chi_1, \chi_2, \ldots, \chi_n]$, define $\Delta(\boldsymbol{\chi}) = \max_{\chi_i, \chi_j \in \boldsymbol{\chi}} d_{\text{TV}}(\chi_i, \chi_j)$, where $d_{\text{TV}}(\chi_i, \chi_j)$ represents the total variation distance between $\chi_i$ and $\chi_j$. The value of $\Delta(\boldsymbol{\chi})$ lies between 0 and 1, with $\Delta(\boldsymbol{\chi}) = 0$ indicating that all distributions in $\boldsymbol{\chi}$ are identical.

## 3 METHOD: INDEX2SORT

In this section, we first describe the proposed Index2Sort algorithm in Section 3.1, then present its complexity theorems in Section 3.2, and finally summarize in Section 3.3 the corollaries obtained by applying our framework to several known indexes.

### 3.1 ALGORITHM OF INDEX2SORT

Index2Sort recursively sorts a portion of the input array, constructs an index using the sorted portion, and then performs bucket sort on the remaining elements of the input array using the constructed index. The algorithm is visualized in Figure 1 and its pseudocode is presented in Algorithm 1. If the length of the input array is smaller than a certain threshold $\tau$, we sort the array using a standard algorithm, such as MergeSort. In the following, let the length of the input array be $n$ ($\geq \tau$) and the input array be $\boldsymbol{x}$ ($\in \mathbb{R}^n$).

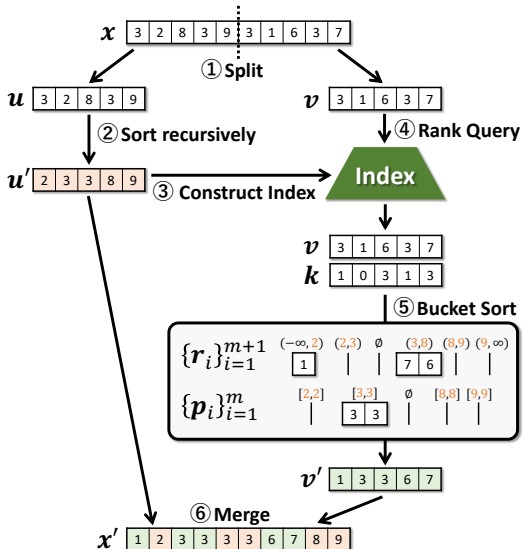

**Algorithm 1** Index2Sort

1: **Input:** $x \in \mathbb{R}^n$ (the array to be sorted)
2: **Output:** $x' \in \mathbb{R}^n$ (the sorted version of $x$)
3: **function** INDEX2SORT($x$)
4:     $n \leftarrow |x|, \; m \leftarrow \lfloor n/2 \rfloor$
5:     **if** $n < \tau$ **then**
6:         **return** MERGESORT($x$)
7:     $u \leftarrow x[1:m], \; v \leftarrow x[m+1:n]$    $\}$ ①
8:     $u' \leftarrow$ INDEX2SORT($u$)               $\}$ ②
9:     $\mathcal{I} \leftarrow$ CONSTRUCTINDEX($u'$)       $\}$ ③
10:     $k \leftarrow []$
11:     **for** $i = 1, \ldots, n - m$           $\}$ ④
12:         $k$.append($\mathcal{I}$.rank($v_i$))
13:     $r_1 \leftarrow [], \ldots, r_{m+1} \leftarrow []$
14:     $p_1 \leftarrow [], \ldots, p_m \leftarrow []$
15:     **for** $i = 1, \ldots, n - m$
16:         **if** $k_i = m \vee v_i \neq u'_{k_i+1}$ **then**
17:             $r_{k_i+1}$.append($v_i$)
18:         **else**                      $\}$ ⑤
19:             $p_{k_i+1}$.append($v_i$)
20:     **for** $i = 1, \ldots, m + 1$
21:         $r'_i \leftarrow$ MERGESORT($r_i$)
22:     $v' \leftarrow$ CONCAT($r'_1, p_1, \ldots, p_m, r'_{m+1}$)
23:     $x' \leftarrow$ MERGE($u', v'$)        $\}$ ⑥
24:     **return** $x'$

Figure 1: Index2Sort algorithm. After recursively sorting a portion of the input array with Index2Sort, an index is constructed using the sorted array, and the remaining elements are then bucket-sorted using this index.

First, the input array $x$ is split into two parts, $u$ and $v$. For theoretical guarantees, $x$ is shuffled once using an $\mathcal{O}(n)$ algorithm, such as the Fisher–Yates shuffle (Fisher & Yates, 1953), before being split into $u$ and $v$. This shuffle is performed only once and is not required during subsequent recursive calls. We define $u = x[1:m]$ and $v = x[m+1:n]$, where $m = \lfloor \alpha n \rfloor$ for an arbitrary constant $\alpha \in (0,1)$. For simplicity, we assume $\alpha = 1/2$, i.e., $m = \lfloor n/2 \rfloor$, in the following explanation. However, the algorithm and its computational guarantees remain valid for any $\alpha \in (0,1)$.

The algorithm then recursively sorts $u$ using Index2Sort. After obtaining the sorted array $u'$, an index is constructed on $u'$. Note that constructing the index requires a sorted array, and $u'$ satisfies this condition. In this way, Index2Sort makes it possible to utilize static indexes for sorting.

Next, the constructed index is used to bucket-sort $v$. For each $v_i \in v$, we perform a rank query on the index, obtaining $k \in \{0, \ldots, m\}^{n-m}$ where $k_i$ is the rank of $v_i$ in $u'$. We then prepare $m + 1$ range buckets $(r_1, \ldots, r_{m+1})$ and $m$ point buckets $(p_1, \ldots, p_m)$ (as detailed in Section 3.2, we introduce these two types of buckets for theoretical guarantees). Each range bucket stores elements that fall within the open intervals between successive elements of $u'$, while each point bucket stores values that exactly match certain elements of $u'$. Concretely, $v_i$ is placed into $r_{k_i+1}$ if $k_i = m$ or $v_i \neq u'_{k_i+1}$; otherwise, it is placed into $p_{k_i+1}$.

Each range bucket is then sorted (e.g., by MergeSort; any $\mathcal{O}(n^2)$ method suffices, as detailed in Appendix A), and the range and point buckets are merged alternately to produce the sorted array $v'$. Finally, $u'$ and $v'$ are merged in the manner of MergeSort to produce the array $x'$, which is the sorted version of $x$.

## 3.2 THEOREMS ON COMPLEXITY OF INDEX2SORT

**Fundamental Theorem.** First, we present the most fundamental and intuitive result, applicable when the complexity guarantees of the index do not rely on distributional assumptions.

**Theorem 3.1.** *Consider a static index algorithm satisfying: (1) given a sorted array of length $n$, the index is constructed in $\mathcal{O}(nC(n))$ expected time; (2) the index answers a rank query in $\mathcal{O}(Q(n))$ expected time. Then, Index2Sort sorts an array of length $n$ in $\mathcal{O}(nC(n) + nQ(n))$ expected time.*

A rigorous proof is given in Appendix A.1; we outline the intuition here. Note that in the following analysis, we expand the recursion performed in the step ② and accumulate the time complexity for each step from ① to ⑥. Steps ① (splitting) and ⑥ (merging) each take $\mathcal{O}(n)$ time. Step ③ constructs indexes for arrays of lengths $n/2$, $n/4$, ... with costs $\mathcal{O}((n/2)C(n/2))$, $\mathcal{O}((n/4)C(n/4))$, ..., summing to $\mathcal{O}(nC(n))$ since $C$ is non-decreasing. Similarly, the total complexity of ④ is $\mathcal{O}(nQ(n))$. Therefore, the only nontrivial part is the total expected time complexity of ⑤. We show that this complexity is $\mathcal{O}(n)$ by adapting a classical probabilistic analysis of bucket size distributions in (Frazer & McKellar, 1970) to our setting. Therefore, the total time complexity of Index2Sort is $\mathcal{O}(nC(n) + nQ(n))$.

We emphasize that point buckets are essential for Index2Sort to achieve the overall complexity of $\mathcal{O}(nC(n) + nQ(n))$. Without them, simply assigning elements to $m+1$ buckets based on rank queries does not guarantee that step ⑤ runs in $\mathcal{O}(n)$ expected time. For example, if a particular value appears $\Omega(n)$ times in the input array $\boldsymbol{x}$, all occurrences fall into the same bucket, requiring $\Omega(n \log n)$ time to sort the bucket. Index2Sort avoids this by using point buckets: for each element, we perform a constant-time check to decide whether it belongs to a range bucket or a point bucket. This keeps each range bucket $\mathcal{O}(1)$ in size with high probability, so sorting them costs $\mathcal{O}(n)$. Since all elements in a point bucket are identical and need not be sorted, the total expected cost of step ⑤ remains $\mathcal{O}(n)$. Consequently, Index2Sort preserves the overall $\mathcal{O}(nC(n) + nQ(n))$ complexity even in the presence of many duplicate elements.

**Under Distributional Assumptions.** Theorem 3.1 cannot be applied directly when the theoretical guarantee of the index relies on assumptions about the distribution of input arrays and queries, which is common in learned indexes. To cover these cases, we provide two companion results: the i.i.d. setting (Theorem 3.2) and the distribution-shift setting (Theorem 3.3).

**Theorem 3.2.** *Consider a static index algorithm satisfying: (1) given a sorted array of length $n$ whose elements are sampled i.i.d. from a distribution $\chi \in \mathfrak{X}$, the index is constructed in $\mathcal{O}(nC(n))$ expected time; (2) given a query independently sampled from the same distribution $\chi$, the index returns the rank of the query in $\mathcal{O}(Q(n))$ expected time. Then, Index2Sort sorts an array of $n$ i.i.d. samples from $\chi \in \mathfrak{X}$ in $\mathcal{O}(nC(n) + nQ(n))$ expected time.*

The proof of this theorem follows almost the same steps as the proof of Theorem 3.1, as detailed in Appendix A.1. It is worth noting that the assumption that each element of $\boldsymbol{x}$ is sampled i.i.d. from the distribution $\chi$ propagates to the elements of $\boldsymbol{u}$ and $\boldsymbol{v}$. This propagation ensures that the complexity of constructing the index on $\boldsymbol{u}'$ is bounded by $\mathcal{O}(nC(n))$ and that the complexity of performing rank queries on all elements of $\boldsymbol{v}$ is bounded by $\mathcal{O}(nQ(n))$.

**Theorem 3.3.** *Consider a static index algorithm satisfying: (1) given a sorted array of $n$ samples from a distribution in $\mathfrak{X}$ with shift at most $\delta$, the index is constructed in $\mathcal{O}(nC(n,\delta))$ expected time; (2) given a query from the same distribution class with shift at most $\delta$, the index returns its rank in $\mathcal{O}(Q(n,\delta))$ expected time. Then, Index2Sort sorts an array sampled from $\mathfrak{X}$ (with $\delta$ distribution shift) in $\mathcal{O}(nC(n,\delta) + nQ(n,\delta))$ expected time.*

The proof of this theorem is similar to that of Theorem 3.2, with the detailed proof given in Appendix A.1. Notably, in the expected time complexity of Index2Sort, $\delta$ appears only in the functions $C$ and $Q$. In other words, the distribution shift impacts only the index construction and query processing steps; the efficiency of the rest of the components of the Index2Sort algorithm is unaffected.

**Handling Approximate Rank Queries.** Furthermore, when the index algorithm supports *approximate rank queries*, which return approximate ranks with a maximum error of $\varepsilon$ instead of exact ranks, the time complexity of Index2Sort can still be guaranteed. This scenario is common because many index structures incorporate mechanisms that provide approximate ranks. These indexes typically refine it to obtain the exact rank through methods such as binary search or exponential search. For example, in a B-tree, each node typically corresponds to a page block that stores multiple data records. As a result, the pure query response of a B-tree has an error bounded by the block size. Similarly, in some learned indexes, such as the PGM-index (Ferragina & Vinciguerra, 2020), the maximum error is explicitly specified as a parameter during the index construction.

The complexity guarantee of Index2Sort under this setting is achieved by making one of the following minor modifications to the algorithm for ⑤: (i) using the sorting algorithm with predictions

| Index | $C(n)$, $Q(n)$ | Assumption | Complexity of Index2Sort |
|---|---|---|---|
| - (Binary Search) | $C(n) = 0$, $Q(n) = \log n$ | - | $\mathcal{O}(n \log n)$ |
| B-tree | $C(n) = Q(n) = \log n$ | - | $\mathcal{O}(n \log n)$ |
| RDA Index | $C(n) = Q(n) = \log \log n$ | $\boldsymbol{D} \overset{\text{iid}}{\sim} \chi \in \mathfrak{X}_{\rho_1, \rho_2}$ | $\mathcal{O}(n \log \log n)$ |
| ESPC Index | $C(n) = Q(n) = 1$ | $\boldsymbol{D} \overset{\text{iid}}{\sim} \chi \in \mathfrak{X}_{\rho_f}$ | $\mathcal{O}(n)$ |
| ESPC Index | $C(n) = 1$, $Q(n) = \log \log n$ | $\boldsymbol{D} \overset{\text{iid}}{\sim} \chi \in \mathfrak{X}_C$ | $\mathcal{O}(n \log \log n)$ |
| Dynamic LI | $C(n) = Q(n) = \log \log n + \log(\delta n)$ | $\boldsymbol{D} \sim \boldsymbol{\chi} \subset \mathfrak{X}_{\rho_1, \rho_2}$ $\wedge \Delta(\boldsymbol{\chi}) \leq \delta$ | $\mathcal{O}(n \log \log n + n \log(\delta n))$ |

Table 1: Computational complexity of Index2Sort using various index structures: RDA Index (Zeighami & Shahabi, 2023), ESPC Index (Croquevielle et al., 2025), and Dynamic LI (Zeighami & Shahabi, 2024).

proposed in (Bai & Coester, 2023) (with slight modifications) to sort $\boldsymbol{v}$ instead of performing bucket sort, or (ii) performing an exponential search on $\boldsymbol{u}'$, starting from the approximate rank to obtain the exact rank. With either modification, the time complexity of Index2Sort can be bounded as follows:

**Theorem 3.4.** *Consider a static index algorithm satisfying: (1) given a sorted array of length $n$, the index is constructed in $\mathcal{O}(nC(n))$ expected time; (2) the index returns an approximate rank with error at most $\varepsilon$ in $\mathcal{O}(Q(n))$ expected time. If the step ⑤ is implemented using either (i) sorting algorithm with predictions (Bai & Coester, 2023), or (ii) exponential search starting from the approximate rank, then Index2Sort sorts in $\mathcal{O}(nC(n) + nQ(n) + n\log(\varepsilon + 1))$ expected time.*

In either case of (i) and (ii), the time complexity of ⑤ is bounded by $\mathcal{O}(n(1+\log(\varepsilon+1)))$. Therefore, as in Theorem 3.1, the overall time complexity of Index2Sort is proved to be $\mathcal{O}(nC(n) + nQ(n) + n\log(\varepsilon + 1))$. The detailed proof is provided in Appendix A.2.

**Worst-Case Complexity.** In addition to the expected time complexity analysis, we also provide the following theorem on the worst-case time complexity of Index2Sort.

**Theorem 3.5.** *Consider a static index algorithm satisfying: (1) given a sorted array of length $n$, the index is constructed in $\mathcal{O}(nC(n))$ worst-case time. (2) the index answers a rank query in $\mathcal{O}(Q(n))$ worst-case time. Also, assume that the algorithm used for sorting each range bucket in the step ⑤ of Index2Sort has a worst-case time complexity of $\mathcal{O}(R(n))$ for sorting an array of length $n$, where $R(n)$ is a superadditive function, i.e., for any $n_1 \geq 0$ and $n_2 \geq 0$, $R(n_1 + n_2) \geq R(n_1) + R(n_2)$. Then, Index2Sort sorts an array of length $n$ in $\mathcal{O}(nC(n) + nQ(n) + R(n))$ worst-case time.*

This theorem implies that in a typical setting, where $C(n) = \log n$, $Q(n) = \log n$, and $R(n) = n \log n$, the worst-case time complexity of Index2Sort is $\mathcal{O}(n \log n)$. This matches the complexity of many classical comparison-based sorting algorithms. A rigorous proof is provided in Appendix A.3.

**Beyond Sorting.** Furthermore, we show that in general contexts beyond sorting, the methods and theoretical results of algorithms with predictions can be extended to a problem setting where only the algorithm for generating predictions is provided. The detailed generalization process and associated theoretical guarantees are provided in Appendix B.

### 3.3 Derived Computational Guarantees for Index2Sort

Here, we summarize the consequences of applying our proposed framework to several classical and learned indexes (Table 1).

**Trivial but Revealing Case.** As the most trivial case, consider answering rank queries using binary search without constructing an index. Here, our Index2Sort closely resembles the existing "Index Sort" (Gurram & Gera, 2011) (note that "Index" here refers to array indices, not an index data structure). However, "Index Sort" does not provide any time complexity guarantees. Since $C(n) = 0$ and $Q(n) = \log n$, Theorem 3.1 shows that its time complexity is $\mathcal{O}(n \log n)$, which is a novel observation. For classical index data structures, such as B-tree, where $C(n) = Q(n) = \log n$, Theorem 3.1 implies that the time complexity of Index2Sort using this index is also $\mathcal{O}(n \log n)$.

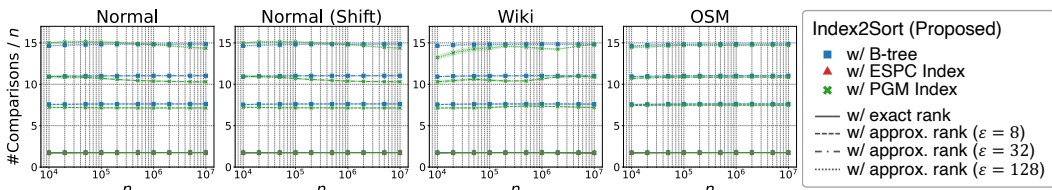

Figure 2: Number of element comparisons required to sort an array of length $n$. Regardless of the distribution, the type of index used, or the precision of rank queries (whether exact or approximate), the number of comparisons required for ⑤ in Index2Sort is observed to be linear with respect to $n$.

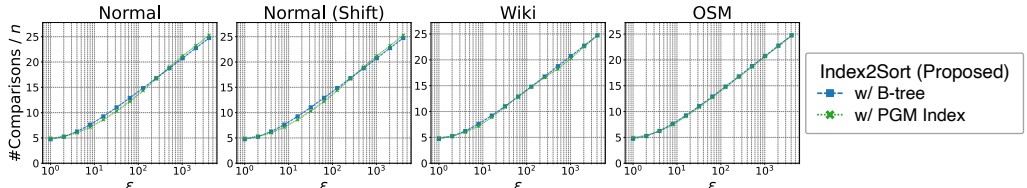

Figure 3: Number of element comparisons in Index2Sort when using an index with a maximum error of $\varepsilon$. Regardless of the distribution or the type of index used, the number of comparisons required for ⑤ in Index2Sort is observed to be proportional to $\log \varepsilon$.

**Structure-Agnostic Proof of $\mathcal{O}(n \log \log n)$ Complexity.** Next, we present the computational guarantees of Index2Sort with learned indexes. The learned index of (Zeighami & Shahabi, 2023) assumes data points are sampled i.i.d. from $\chi \in \mathfrak{X}_{\rho_1, \rho_2}$ and achieves $C(n) = Q(n) = \log \log n$. By Theorem 3.2, the time complexity of Index2Sort with this learned index is $\mathcal{O}(n \log \log n)$. This is equivalent to the guarantees in (Sato & Matsui, 2024; Zeighami & Shahabi, 2024), but our Index2Sort achieves the same result without requiring any observation of the internal structure of the learned index, making both the algorithm and its time complexity guarantees more intuitive.

**From $\mathcal{O}(n \log \log n)$ to $\mathcal{O}(n)$.** Index2Sort using ESPC-index (Croquevielle et al., 2025) offers stronger theoretical guarantees than any existing learned sort. When the data and queries are independently sampled from $\chi \in \mathfrak{X}_{\rho_f}$, ESPC-index achieves $C(n) = Q(n) = 1$. By Theorem 3.2, this gives Index2Sort an expected time complexity of $\mathcal{O}(n)$ under $\chi \in \mathfrak{X}_{\rho_f}$. This is a **tighter** guarantee under **weaker** assumptions than prior learned sorts (Sato & Matsui, 2024; Zeighami & Shahabi, 2024), which achieve $\mathcal{O}(n \log \log n)$ under $\chi \in \mathfrak{X}_{\rho_1, \rho_2}$.

$\mathcal{O}(n \log \log n)$ **under the Weakest Assumptions.** Moreover, using ESPC allows Index2Sort to obtain strong expected complexity guarantees even under very weak distributional assumptions. When the data and queries are independently sampled from $\chi \in \mathfrak{X}_C$, ESPC-index achieves $C(n) = 1$ and $Q(n) = \log \log n$ (this result is not mentioned in the original paper, but we prove it in Appendix C). Therefore, by Theorem 3.2, the expected time complexity of Index2Sort is $\mathcal{O}(n \log \log n)$ under $\chi \in \mathfrak{X}_C$. This is the first theoretical guarantee for learned sorts under the very weak distributional assumption $\chi \in \mathfrak{X}_C$.

**Complexity Guarantees under Distribution Drift.** Finally, the learned index of (Zeighami & Shahabi, 2024) assumes $D \sim \chi \subset \mathfrak{X}_{\rho_1, \rho_2}$ and $\Delta(\chi) \leq \delta$, yielding $C(n, \delta) = Q(n, \delta) = \log \log n + \log(\delta n)$. Thus, by Theorem 3.3, Index2Sort runs in $\mathcal{O}(n \log \log n + n \log(\delta n))$ expected time. Although this complexity guarantee is not novel because the index also supports insertion, we include this case to illustrate how Index2Sort inherits guarantees even under distribution shift. Future static learned indexes with theoretical guarantees under distribution shifts could be seamlessly incorporated into our framework in the same manner.

## 4 EXPERIMENTS

In this section, we experimentally validate the complexity results in our theorems, focusing on step ⑤ of Index2Sort, where the array $v$ is sorted using the index output. This focus is because, as noted in the intuitive proof of Theorem 3.1, the complexities of other steps are obvious. The only non-trivial points are: (1) if the index provides exact ranks, the complexity of ⑤ is $\mathcal{O}(n)$ (Theorems 3.1 to 3.3), and (2) if the index provides approximate ranks, the complexity of ⑤ is $\mathcal{O}(n(1 + \log(\varepsilon + 1)))$ (Theorem 3.4). We evaluate this complexity under various distributions and indexes, including the cumulative complexity arising from recursive calls in ②.

**Setup.** We used both artificial data and real-world data to support our theorem. For artificial data, we considered two distributions: **Normal**, where each element is drawn from $\mathcal{N}(0, 1)$; and **Normal (Shift)**, a distribution with a linearly shifting mean; the $i$-th element of the input array is drawn from $\mathcal{N}(i/n, 1)$. For real-world data, we used **Wiki** (Marcus et al., 2020a), the timestamps of Wikipedia article edits, and **OSM** (Marcus et al., 2020a), OpenStreetMap locations represented as Google S2 CellIds. Input arrays were generated by randomly sampling $n$ elements from these datasets.

All experiments were implemented in C++ and conducted on a single thread on a Linux machine equipped with an Intel® Core™ i9-11900H CPU @ 2.50 GHz and 64 GB of memory. The code was compiled using GCC version 9.4.0 with the `-O3` optimization flag. We set $\tau = 128$ and $\alpha = 1/2$. We report the mean and standard deviation over 10 runs for each data point in the figures. Due to space constraints, we present only a subset of representative results here. For the full set of experiments, covering 24 datasets (synthetic and real-world), 5 index algorithms, and wall-clock comparisons against 10 baseline algorithms, please refer to Appendix D.

**Linearity in** $n$**.** We experimentally show that the number of comparisons in ⑤ of Index2Sort grows linearly with the input array length $n$ under various conditions. Figure 2 plots the number of comparisons in ⑤ against $n$. Here, the approximate rank was handled using the method using exponential search. The results confirm linear growth in $n$, regardless of the distribution, distribution shifts, index type, or rank precision (exact or approximate), supporting Theorems 3.1 to 3.4.

**Proportionality to** $\log \varepsilon$**.** We also show that the number of comparisons in ⑤ scales proportionally to $\log \varepsilon$. Figure 3 shows the number of comparisons in ⑤ when using a B-tree or PGM-index with a maximum error of $\varepsilon$. Here, we set the length of the array to $n = 10^7$. The approximate rank was handled using the method using exponential search. The results confirm proportionality to $\log \varepsilon$, regardless of the distribution, distribution shifts, or index type, supporting Theorem 3.4.

## 5 RELATED WORK

Here, we first give an overview of indexes and sorting methods, focusing on learned indexes and learned sorts in Section 5.1. Then, in Section 5.2, we introduce algorithms with predictions, a closely related field, and discuss its connections and differences with our work.

### 5.1 LEARNED INDEX AND LEARNED SORT

An index, in a broader sense, is a data structure designed to enable fast data access. Examples include B-trees (Bayer & McCreight, 1972), hash maps (Knuth, 1998), and Bloom filters (Bloom, 1970), which are widely used in applications such as databases (Ramakrishnan & Gehrke, 2002), search engines (Schütze et al., 2008), and file systems (Ghemawat et al., 2003). Recently, *learned indexes* have been proposed (Kraska et al., 2018), replacing or augmenting classical structures with machine learning models to improve memory efficiency and query speed. Research has explored machine learning-augmented versions of various data structures, including Bloom filters (Mitzenmacher, 2018; Dai & Shrivastava, 2020; Vaidya et al., 2021; Sato & Matsui, 2023), R-trees (Gu et al., 2023; Abdullah-Al-Mamun et al., 2022), and count-min sketches (Hsu et al., 2019; Zhang et al., 2020; Dolera et al., 2023). In particular, learned indexes with functionality similar to B-tree have been extensively studied (Galakatos et al., 2019; Ferragina & Vinciguerra, 2020; Sun et al., 2023) and are often referred to as learned indexes in the narrow sense. These approaches use machine learning models to approximate the cumulative density function (CDF) of the input array

distribution, enabling better memory efficiency and faster search. Most learned indexes employ a hierarchical structure of linear models (Galakatos et al., 2019; Ferragina & Vinciguerra, 2020; Ding et al., 2020; Wang et al., 2020; Hadian & Heinis, 2020; Li et al., 2021), though other designs, such as those based on polynomial functions (Wu et al., 2021) or neural networks (Kraska et al., 2018), have also been proposed. More recently, there has been increasing interest in learned indexes with theoretical guarantees. Details of these guarantees are discussed in Section 3.3.

Sorting is one of the most fundamental problems in computer science, and a variety of algorithms have been proposed to address it. Comparison-based sorting algorithms, such as Quicksort and Mergesort, have a well-known worst-case complexity of $\Omega(n \log n)$. On the other hand, by using additional information or imposing certain constraints, it is possible to achieve lower worst-case complexity. For instance, RadixSort achieves a worst-case complexity of $\mathcal{O}(nw)$, where $w$ is the number of digits per element. For integer arrays, deterministic algorithms with a complexity of $\mathcal{O}(n \log \log n)$ (Han, 2002) and randomized algorithms with an expected complexity of $\mathcal{O}(n\sqrt{\log \log n})$ (Han & Thorup, 2002) have been proposed. For real-valued arrays, a recent algorithm achieves a complexity of $\mathcal{O}(n\sqrt{\log n})$ (Han, 2020). Inspired by learned indexes, sorting algorithms using machine learning models to approximate the CDF, referred to as *learned sort*, have been proposed (Kraska et al., 2019). The learned sort algorithms perform sorting quickly by efficiently assigning keys to buckets using the predicted CDF and reducing comparisons (Kristo et al., 2020; 2021). More recently, by redesigning the architecture of a learned index for sorting, a learned sort with $\mathcal{O}(n \log \log n)$ expected complexity under the assumption that $D \overset{\text{iid}}{\sim} \chi \in \mathfrak{X}_{\rho_1, \rho_2}$ is introduced (Sato & Matsui, 2024; Zeighami & Shahabi, 2024). In contrast, our Index2Sort adopts a different approach by treating the index as an opaque box, achieving stronger guarantees.

## 5.2 Algorithms with predictions

Algorithms with predictions (Mitzenmacher & Vassilvitskii, 2022) is a rapidly growing field that has received considerable attention in recent years. These studies have shown that when predictions are accurate, performance can significantly exceed that of algorithms without predictions, while maintaining robust performance even when predictions are inaccurate or adversarial. Early research in this area focused primarily on classic online problems, such as caching (Narayanan et al., 2018; Rohatgi, 2020; Lykouris & Vassilvitskii, 2021; Antoniadis et al., 2023b; Sadek & Elias, 2024), rent-or-buy problems (Purohit et al., 2018; Gollapudi & Panigrahi, 2019; Shin et al., 2023), and scheduling (Mitzenmacher, 2020; Lattanzi et al., 2020; Lassota et al., 2023; Elias et al., 2024). The scope of these techniques has been extended to offline problems, including matching (Dinitz et al., 2021; Sakaue & Oki, 2022; Choo et al., 2024), clustering (Ergun et al., 2022; Nguyen et al., 2023), and graph algorithms (Chen et al., 2022; Davies et al., 2023; Polak & Zub, 2024). There has also been significant progress in sorting with predictions (Lu et al., 2021; Chan et al., 2023; Erlebach et al., 2023). In particular, (Bai & Coester, 2023) proposed a generalized sorting algorithm with predictions that offer tight complexity guarantees.

While many studies assume that predictions are passively obtained at no cost, while others focus on optimizing the predictions themselves. For example, there are studies that propose algorithms to reduce the number of predictions used (Im et al., 2022; Drygala et al., 2023; Benomar & Perchet, 2023; Aamand et al., 2023; Sadek & Elias, 2024), and some limit the size per prediction (Mitzenmacher, 2021; Dütting et al., 2021; Antoniadis et al., 2023a). In addition, research efforts have been made to design customized loss functions for training machine learning models used to generate predictions (Du et al., 2021; Anand et al., 2020) or to train machine learning models dynamically using online-learning methods (Khodak et al., 2022; Sakaue & Oki, 2022; 2023). These studies share a common direction in that they refine the predictions themselves.

While our work shares similarities with these approaches, it fundamentally differs in that we include the training time of the machine learning model as part of the computational cost. We introduce a new problem setting that explicitly accounts for both training and inference costs. This perspective is particularly crucial for end-to-end performance analysis in offline problems, where predictions are tailored to each individual problem instance.

## 6 Limitations and Future Work

While our analysis provides strong guarantees in terms of expected running time, the worst-case complexity of Index2Sort is $\mathcal{O}(nC(n) + nQ(n) + R(n))$, where $R(n)$ is the worst-case complexity of the sorting algorithm applied to each range bucket. In practice, this still provides strong protection against slowdowns: under typical settings, it simplifies to $\mathcal{O}(n \log n)$, matching the bounds for classical comparison-based sorting and preventing catastrophic performance degradation. Nonetheless, the explicit dependence on $R(n)$ points to a natural direction for future work: can this dependence be removed through a more refined analysis or alternative algorithmic designs? Eliminating it could yield tighter worst-case guarantees and further strengthen the theoretical foundation of Index2Sort.

Another interesting open problem is developing a *Sort2Index* framework. Specifically, given a sorting algorithm, can we construct an indexing algorithm based on that sorting algorithm and provide theoretical guarantees on its computational complexity? Although the equivalence between sorting and priority queues has been established (Thorup, 2007), two key differences between priority queues and static indexes make this problem worth investigating: (1) priority queues are dynamic, while static indexes are static, and (2) priority queues support only minimum value extraction operations, while static indexes answer rank queries.

Finally, we note that our contributions are primarily theoretical. Although Index2Sort provides stronger asymptotic guarantees than existing sorting algorithms, it does not necessarily outperform them in practical runtime due to the lack of low-level hardware optimizations (see Appendix D). Bridging this theory-practice gap through hardware-conscious design or implementation-level optimizations is an important direction for future work.

## 7 Conclusion

In this paper, we proposed Index2Sort, a general framework for deriving sorting algorithms from static indexes. We proved that Index2Sort automatically inherits the computational guarantees of the underlying index, yielding strictly stronger complexity bounds than existing learned sorts. This work bridges the gap between theory on learned indexes and learned sorts, enabling future advances in index research to be transferred directly to sorting.

**Reproducibility Statement** Our theoretical results are accompanied by clear descriptions of all assumptions and complete proofs, provided in Appendix A. The datasets used in our experiments, parameter settings, and computational environment are thoroughly described in Section 4 and Appendix D. The code used for our experiments is submitted as supplementary material and will be made publicly available on GitHub upon acceptance.

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

# A PROOFS

Here, we provide the proofs omitted in the main text. First, in Appendix A.1, we present the proofs of the three fundamental theorems of Index2Sort: Theorems 3.1 to 3.3. Next, in Appendix A.2, we detail the necessary modifications to the Index2Sort algorithm for handling approximate rank queries and prove the corresponding time complexity guarantee stated in Theorem 3.4. Finally, in Appendix A.3, we introduce and prove Theorem 3.5, which establishes the worst-case time complexity of Index2Sort. While the main text explains the algorithm assuming $\alpha = 1/2$ for simplicity, in the following, we generalize the analysis to allow the number of buckets, $m$, to be defined as $m = \lfloor \alpha n \rfloor$ for any constant $\alpha \in (0, 1)$. Additionally, while the main text describes MergeSort as the algorithm used for sorting range buckets, in the following, we allow any sorting algorithm with a time complexity of $\mathcal{O}(n^2)$. All expectations and probabilities in our analysis are computed with respect to the random shuffle performed by the Index2Sort algorithm (①). Additionally, if the input array or query is assumed to be drawn from a certain distribution, the corresponding sampling randomness is also included in our analysis.

## A.1 PROOF OF THEOREMS 3.1 TO 3.3

Here, we first present Theorem A.1, a lemma that provides the theoretical guarantees for the bucket sorting step in Index2Sort. We then use this lemma to prove the theorems Theorems 3.1 to 3.3.

**Lemma A.1.** *The expected time complexity for the step ⑤ of Index2Sort is $\mathcal{O}(n)$.*

*Proof of Theorem A.1.* Let $A_i$ ($i \in \{1, 2, \ldots, m+1\}$) denote the number of elements in the $i$-th range bucket when Index2Sort is applied to an input array of length $n$. Specifically, $A_i$ denotes the number of elements in the $i$-th range bucket $r_i$, obtained by bucketing the array $v \in \mathbb{R}^{n-m}$ using the thresholds $u \in \mathbb{R}^m$ while using the point-bucket mechanism.

Additionally, let $B_i$ ($i \in \{1, 2, \ldots, m+1\}$) denote the number of elements in the $i$-th bucket obtained by bucketing the same $v \in \mathbb{R}^{n-m}$ using the same thresholds $u \in \mathbb{R}^m$ **without** applying the point bucket mechanism; that is, each element is always assigned to exactly one of the $m+1$ buckets. Here, to disambiguate the handling of values equal to any threshold in $u$, we associate each input element $x_i$ with its original index $i$, forming tuples $(x_i, i)$. These tuples are then totally ordered, so ties in value are resolved by input order. Bucketing is performed straightforwardly according to this total order, allowing us to utilize the results of (Frazer & McKellar, 1970) for analysis of $B$.

Then, for the same $u$ and $v$, $A_i \leq B_i$ holds for all $i$. This is because if $v_j$ ($j \in \{1, 2, \ldots, n-m\}$) falls into the $i$-th range bucket $r_i$ using the Index2Sort method, then $v_j$ will also fall into the $i$-th bucket in the bucketing procedure described in the definition of $B$. Therefore,

$$\mathbb{E}\left[\sum_{i=1}^{m+1} A_i^2\right] = \sum_{i=1}^{m+1} \mathbb{E}\left[A_i^2\right] \tag{1}$$

$$\leq \sum_{i=1}^{m+1} \mathbb{E}\left[B_i^2\right]. \tag{2}$$

Now, from Lemma 1 in (Frazer & McKellar, 1970), $\Pr[B_i = j] = \binom{n-j-1}{m-1}/\binom{n}{m}$. Therefore,

$$\sum_{i=1}^{m+1} E\left[B_i^2\right] = \sum_{i=1}^{m+1}\sum_{j=0}^{n-m} j^2 \Pr\left[B_i = j\right] \tag{3}$$

$$= \sum_{i=1}^{m+1}\sum_{j=0}^{n-m} j^2 \cdot \frac{\binom{n-j-1}{m-1}}{\binom{n}{m}} \tag{4}$$

$$= \frac{m+1}{\binom{n}{m}} \sum_{j=0}^{n-m} j^2 \binom{n-j-1}{m-1}. \tag{5}$$

Here, we evaluate the sum as follows:

$$\sum_{j=0}^{n-m} j^2 \binom{n-j-1}{m-1} \tag{6}$$

$$= \sum_{k=m-1}^{n-1} (n-1-k)^2 \binom{k}{m-1} \quad (k := n-j-1) \tag{7}$$

$$= (n-1)^2 \sum_{k=m-1}^{n-1} \binom{k}{m-1} - 2(n-1) \sum_{k=m-1}^{n-1} k \binom{k}{m-1} + \sum_{k=m-1}^{n-1} k^2 \binom{k}{m-1} \tag{8}$$

$$= (n-1)^2 \binom{n}{m} - 2(n-1)\left( (m-1)\binom{n}{m} + m\binom{n}{m+1} \right) + \left( (m-1)^2 \binom{n}{m} \right. \tag{9}$$

$$\left. + m(2m-1)\binom{n}{m+1} + m(m+1)\binom{n}{m+2} \right) \quad (\because \text{Hockey-stick identity}) \tag{10}$$

$$= (n-m)^2 \binom{n}{m} + m(2m-2n+1)\binom{n}{m+1} + m(m+1)\binom{n}{m+2} \tag{11}$$

$$= \left( (n-m)^2 + m(2m-2n+1)\cdot\frac{n-m}{m+1} + m(m+1)\cdot\frac{(n-m)(n-m-1)}{(m+2)(m+1)} \right) \binom{n}{m} \tag{12}$$

$$= \frac{(n-m)(2n-m)}{(m+1)(m+2)} \binom{n}{m}. \tag{13}$$

Therefore,

$$\sum_{i=1}^{m+1} E\left[ B_i{}^2 \right] = \frac{m+1}{\binom{n}{m}} \cdot \frac{(n-m)(2n-m)}{(m+1)(m+2)} \binom{n}{m} \tag{14}$$

$$= \frac{(n-m)(2n-m)}{m+2}. \tag{15}$$

With $m = \lfloor \alpha n \rceil$ for a fixed $\alpha \in (0,1)$, this expression is $\Theta(n)$. Therefore,

$$\mathbb{E}\left[ \sum_{i=1}^{m+1} A_i{}^2 \right] = \mathcal{O}(n). \tag{16}$$

Therefore, since Index2Sort uses a sorting algorithm with $\mathcal{O}(n^2)$ time complexity for sorting each range bucket, the expected time complexity for the step ⑤ is $\mathcal{O}(n)$. □

Using this lemma, we provide the proofs for Theorems 3.1 to 3.3.

*Proof of Theorem 3.1.* Let the expected time complexity of Index2Sort be $S(n)$. Using mathematical induction, we show that $S(n) = \mathcal{O}(nC(n) + nQ(n) + n)$. The time complexity of each step in the Index2Sort algorithm is as follows:

① Splitting the array requires $\mathcal{O}(n)$ computations. The initial shuffle is also $\mathcal{O}(n)$.

② Recursively sorting $\boldsymbol{u}$ requires $S(\alpha n)$.

③ Constructing the index on $\boldsymbol{u}$ has a expected complexity of $\mathcal{O}(\alpha n C(\alpha n))$.

④ Answering rank queries for all elements of $\boldsymbol{v}$ using the index requires $\mathcal{O}((1-\alpha)nQ(\alpha n))$ expected complexity.

⑤ Sorting $\boldsymbol{v}$ using the results of rank queries has an expected complexity of $\mathcal{O}(n)$ (by Theorem A.1).

⑥ Merging $\boldsymbol{u'}$ and $\boldsymbol{v'}$ requires $\mathcal{O}(n)$.

For ⑤, we rely on Theorem A.1 and the assumption that the sorting algorithm used for range buckets has a complexity of $\mathcal{O}(n^2)$. Thus, $S(n)$ can be expressed recursively as follows:

$$S(n) = S(\alpha n) + \mathcal{O}(\alpha n C(\alpha n) + (1 - \alpha)nQ(\alpha n) + n). \tag{17}$$

Here, suppose there exist constants $c \in \mathbb{R}_{>0}$ and $n_0 \in \mathbb{N}$ such that for any $n_0 \leq n' < n$, we have:

$$S(n') \leq c(n'C(\alpha n') + n'Q(\alpha n') + n'). \tag{18}$$

In the following, we show that by taking a sufficiently large constant $c$ (which does not depend on $n$), we can obtain $S(n) \leq c(nC(\alpha n) + nQ(\alpha n) + n)$. We consider two cases: when $\alpha n \geq n_0$ and when $\alpha n < n_0$.

In the first case, i.e., when $\alpha n \geq n_0$, from Equation (18), it follows that

$$S(\alpha n) \leq c(\alpha n C(\alpha^2 n) + \alpha n Q(\alpha^2 n) + \alpha n). \tag{19}$$

Therefore, from Equation (17), we get

$$S(n) \leq c(\alpha n C(\alpha^2 n) + \alpha n Q(\alpha^2 n) + \alpha n) + \mathcal{O}(\alpha n C(\alpha n) + (1 - \alpha)nQ(\alpha n) + n). \tag{20}$$

Here, by defining $\beta := \max(\alpha, 1 - \alpha)$ and taking $c$ sufficiently large, we can rewrite this as:

$$S(n) \tag{21}$$
$$\leq c(\beta n C(\alpha^2 n) + \beta n Q(\alpha^2 n) + \beta n) + \mathcal{O}(\beta n C(\alpha n) + \beta n Q(\alpha n) + n) \tag{22}$$
$$\leq c(\beta n C(\alpha n) + \beta n Q(\alpha n) + \beta n) + \mathcal{O}(n C(\alpha n) + n Q(\alpha n) + n) \tag{23}$$
$$= c(n C(\alpha n) + n Q(\alpha n) + n) - c(1 - \beta)(n C(\alpha n) + n Q(\alpha n) + n) + \mathcal{O}(n C(\alpha n) + n Q(\alpha n) + n) \tag{24}$$
$$\leq c(n C(\alpha n) + n Q(\alpha n) + n). \tag{25}$$

In the second inequality, we use the fact that $C$ and $Q$ are non-decreasing functions of $n$. The final inequality holds by choosing $c$ as a sufficiently large constant.

In the second case, i.e., $\alpha n < n_0$, there exists a certain constant $d > 0$, which does not depend on $n$, such that

$$S(\alpha n) \leq d. \tag{26}$$

This is because, since $\alpha n < n_0$, $S(\alpha n)$ is at most $\max_{n' \in \{1,\dots,n_0-1\}} S(n')$, which does not depend on $n$. Therefore, from Equation (17),

$$S(n) \leq d + \mathcal{O}(\alpha n C(\alpha n) + (1 - \alpha)nQ(\alpha n) + n). \tag{27}$$

Since $n C(\alpha n) + n Q(\alpha n) + n \geq 1$, by taking $c$ sufficiently large,

$$d + \mathcal{O}(\alpha n C(\alpha n) + (1 - \alpha)nQ(\alpha n) + n) \leq c(n C(\alpha n) + n Q(\alpha n) + n). \tag{28}$$

Therefore, from Equations (27) and (28), we get $S(n) \leq c(nC(\alpha n) + nQ(\alpha n) + n)$.

By mathematical induction, we conclude that for any $n \geq n_0$, $S(n) \leq c(nC(\alpha n) + nQ(\alpha n) + n)$. Thus, we have $S(n) = \mathcal{O}(nC(\alpha n) + nQ(\alpha n) + n)$. Since $C$ and $Q$ are non-decreasing functions and $Q(n) \geq 1$, we deduce that $S(n) = \mathcal{O}(nC(n) + nQ(n))$. □

*Proof of Theorem 3.2.* Under the assumption that each element of the input array is sampled i.i.d. from a single distribution $\chi \in \mathfrak{X}$, let the expected time complexity of Index2Sort be $S(n)$. Following an approach similar to the proof of Theorem 3.1, the time complexity of each step in the Index2Sort algorithm is as follows:

① As in Theorem 3.1, the complexity is $\mathcal{O}(n)$.

② Sorting $\boldsymbol{u}$ recursively takes $S(\alpha n)$, since the elements of $\boldsymbol{u}$ are sampled i.i.d. from the same distribution $\chi$.

③ Constructing the index on $\boldsymbol{u}$ has a complexity of $\mathcal{O}(\alpha n C(\alpha n))$, since $\boldsymbol{u}'$ is a sorted version of $\boldsymbol{u}$, an array sampled i.i.d. from the distribution $\chi$.

④ Answering rank queries for all elements of $v$ using the index requires $\mathcal{O}((1-\alpha)nQ(\alpha n))$, since $u'$ is a sorted version of $u$, an array sampled i.i.d. from the distribution $\chi$, and $v$ is also independently sampled from $\chi$.

⑤ As in Theorem 3.1, the complexity is $\mathcal{O}(n)$.

⑥ As in Theorem 3.1, the complexity is $\mathcal{O}(n)$.

For the steps ②, ③, and ④, the propagation of the i.i.d. assumption from $x$ to $u$ and $v$ is utilized. Specifically, the assumption that each element of $x$ is sampled i.i.d. from $\chi \in \mathfrak{X}$ ensures that each element of $u$ and $v$ is also sampled i.i.d. from $\chi \in \mathfrak{X}$. This satisfies the assumptions required for the time complexity guarantees of both Index2Sort and the index, allowing the respective guarantees to be applied. For the step ⑤, since Theorem A.1 does not rely on any distributional assumptions, the complexity remains the same as in Theorem 3.1. Therefore, $S(n)$ can be expressed using the same recurrence relation as in Theorem 3.1 (Equation (17)), leading to the same result, $S(n) = \mathcal{O}(nC(n) + nQ(n))$. □

*Proof of Theorem 3.3.* Under the assumption that each element of the input array is independently sampled from a sequence of distributions $\chi$, where $\chi$ has at most $\delta$ distribution shift and $\chi \subset \mathfrak{X}$, let the expected time complexity of Index2Sort be $S(n)$. Following an approach similar to the proof of Theorem 3.1, the time complexity of each step in the Index2Sort algorithm is as follows:

① As in Theorem 3.1, the complexity is $\mathcal{O}(n)$.

② Sorting $u$ recursively takes $S(\alpha n)$ since the elements of $u$ are independently sampled from a sequence of distributions with at most $\delta$ distribution shift.

③ Constructing the index on $u$ has a complexity of $\mathcal{O}(\alpha n C(\alpha n, \delta))$, since $u'$ is a sorted version of $u$, which is sampled independently from a sequence of distributions with at most $\delta$ distribution shift.

④ Answering rank queries for all elements of $v$ using the index requires $\mathcal{O}((1-\alpha)nQ(\alpha n, \delta))$, because $u'$ is a sorted version of $u$, which is sampled independently from a sequence of distributions with at most $\delta$ distribution shift, and $v$ is also sampled independently from distributions in $\chi$ with at most $\delta$ distribution shift.

⑤ As in Theorem 3.1, the complexity is $\mathcal{O}(n)$.

⑥ As in Theorem 3.1, the complexity is $\mathcal{O}(n)$.

For the steps ②, ③, and ④, the assumption that the elements of $x$ are independently sampled from a sequence of distributions $\chi$ with at most $\delta$ distribution shift ensures that the elements of $u$ and $v$ also follow the same assumption. This allows the time complexity guarantees of both Index2Sort and the index to be applied recursively. For the step ⑤, since Theorem A.1 does not rely on any distributional assumptions, the complexity remains the same as in Theorem 3.1. Thus, $S(n)$ can be expressed as follows:

$$S(n) = S(\alpha n) + \mathcal{O}(\alpha n C(\alpha n, \delta) + (1-\alpha)nQ(\alpha n, \delta) + n), \qquad (29)$$

leading to the result, $S(n) = \mathcal{O}(nC(n, \delta) + nQ(n, \delta))$. □

## A.2 PROOF OF THEOREM 3.4

Next, we prove Theorem 3.4, which demonstrates that Index2Sort remains valid even under the condition that approximate rank queries are allowed. For the guarantees on time complexity, it is necessary to implement one of the two algorithmic modifications mentioned in the main text; (i) Instead of bucket sorting, use a slightly modified version of the sorting with predictions algorithm (Bai & Coester, 2023) to sort $v$, or (ii) Perform an exponential search on $u'$ to determine the exact rank using the approximate rank query result as the starting point. Here, we first present two key lemmas, Theorem A.2 and Theorem A.3, which are critical for guaranteeing the time complexity when modification of (i) is applied. We then show that regardless of whether modification (i) or (ii) is applied, the time complexity of Index2Sort is bounded as stated in Theorem 3.4.

**Modified Displacement Sort**    Displacement Sort, the sorting with predictions algorithm proposed in (Bai & Coester, 2023), is a simple yet effective approach that proceeds as follows:

1. Assign each element to the bucket according to the prediction, which is the predicted position in the sorted array.

2. Insert elements from buckets with smaller predicted values sequentially into a data structure called a finger tree (Guibas et al., 1977).

3. Extract values from the finger tree in increasing order to obtain the sorted array.

A finger tree is a binary tree with a "finger," a pointer to the most recently accessed or inserted element. This structure enables fast access and insertion of elements near the finger. Specifically, accessing or inserting an element at a distance $d$ from the finger can be done in $\mathcal{O}(\log d)$ time. In (Bai & Coester, 2023), this property is leveraged to achieve very low time complexity when the predictions are reasonably accurate.

To make this algorithm applicable to Index2Sort, we introduce the following two minor modifications:

- Extension of the prediction range: In the original algorithm, the prediction range was defined as $\{1, 2, \ldots, l\}$ for an input array of length $l$. We extend this range to a contiguous set of $\Theta(l)$ integers (in our case, the prediction is the approximate rank, so it is in $\{0, 1, 2, \ldots, m\}$). Concretely, we prepare buckets corresponding to each of these $\Theta(l)$ integers and assign elements to buckets based on their predicted values.

- Modification for duplicate handling: Instead of storing only the values in each node of the finger tree, we modify the structure to store both the value and its frequency (i.e., the number of times it has appeared). When inserting a value into the finger tree, if the value already exists, we simply increment its frequency instead of adding a new node.

The first modification is necessary in Index2Sort because the size of the prediction range, $m + 1$, does not necessarily match the length of the array to be sorted, $n - m$. The second modification is required because the original Displacement Sort algorithm assumes there are no duplicate elements in the input array, whereas Index2Sort considers the possibility of duplicate elements.

Now, we provide a theoretical guarantee for the extended Displacement Sort algorithm described above. Consider an input array $\boldsymbol{v} \in \mathbb{R}^l$ of length $l$ with predictions $\hat{\boldsymbol{p}} \in \{1, 2, \ldots, m + 1\}^l$ (where $m = \Theta(l)$). Define the prediction error metric $\eta_i \in \mathbb{N}$ for each element $x_i$ as follows: $\eta_i = |\{v_j \mid j \in \{1, 2, \ldots, l\} \wedge v_i \leq v_j \wedge \hat{p}_j \leq \hat{p}_i\}|$. Then, the following lemma holds:

**Lemma A.2.** *The time complexity of the Displacement Sort algorithm, extended as described above, for sorting the array $\boldsymbol{v}$ is $\mathcal{O}(l + \sum_{i=1}^{l} \log(\eta_i + 1))$.*

*Proof of Theorem A.2.* First, the computational cost of distributing $\boldsymbol{v}$ into buckets using predicted values is $\mathcal{O}(l)$. This is because the number of buckets is $m + 1 = \Theta(l)$, and assigning each element to a bucket takes $\mathcal{O}(1)$.

Next, consider the time complexity of inserting elements into the finger tree. Let the concatenated array of the distributed buckets be $\boldsymbol{w} \in \mathbb{R}^l$. Define $d_i$ ($i \in \{2, 3, \ldots, l\}$) as the number of unique elements in $\{w_1, \ldots, w_{i-1}\}$ that fall within the closed interval of $[w_{i-1}, w_i]$ or $[w_i, w_{i-1}]$, i.e.,

$$d_i = |\{w_j \mid j \in \{1, 2, \ldots, i-1\} \wedge (w_j \in [w_{i-1}, w_i] \vee w_j \in [w_i, w_{i-1}])\}|. \tag{30}$$

When inserting $w_i$ ($i \in \{2, 3, \ldots, l\}$) into the finger tree, the computational cost is $\mathcal{O}(\log d_i)$. This follows from the properties of the finger tree and the fact that only unique elements are inserted into the finger tree, thanks to our extensions. Thus, the total computational cost is $\mathcal{O}(\sum_{i=2}^{l} \log d_i)$. The value of $d_i$ is bounded as follows:

$$d_i = |\{w_j \mid j \in \{1, 2, \ldots, i-1\} \wedge (w_j \in [w_{i-1}, w_i] \vee w_j \in [w_i, w_{i-1}])\}| \tag{31}$$

$$\leq |\{w_j \mid j \in \{1, 2, \ldots, i-1\} \wedge (w_{i-1} \leq w_j \vee w_i \leq w_j)\}| \tag{32}$$

$$\leq |\{w_j \mid j \in \{1, 2, \ldots, i-1\} \wedge w_{i-1} \leq w_j\}| + |\{w_j \mid j \in \{1, 2, \ldots, i-1\} \wedge w_i \leq w_j\}| \tag{33}$$

$$\leq |\{w_j \mid j \in \{1, 2, \ldots, i-2\} \wedge w_{i-1} \leq w_j\}| + |\{w_j \mid j \in \{1, 2, \ldots, i-1\} \wedge w_i \leq w_j\}| + 1. \tag{34}$$

Now, from the definition of $\eta_i$,

$$|\{w_j \mid j \in \{1, 2, \ldots, i-1\} \wedge w_i \leq w_j\}| \leq |\{v_j \mid j \in \{1, 2, \ldots, l\} \wedge v_i \leq v_j \wedge \hat{p}_j \leq \hat{p}_i\}| \quad (35)$$
$$= \eta_i. \quad (36)$$

Therefore, we have

$$d_i \leq \eta_{i-1} + \eta_i + 1. \quad (37)$$

The total computational cost of inserting elements into the finger tree is then:

$$\sum_{i=2}^{l} \mathcal{O}(\log d_i) \leq \sum_{i=2}^{l} \mathcal{O}(\log(\eta_{i-1} + \eta_i + 1)) \quad (38)$$

$$\leq \sum_{i=1}^{l} \mathcal{O}(\log(\eta_i + 1)). \quad (39)$$

Finally, extracting elements from the finger tree in sorted order takes at most $\mathcal{O}(l)$. Thus, the total time complexity of the modified Displacement Sort algorithm for sorting $v$ is $\mathcal{O}(l + \sum_{i=1}^{l} \log(\eta_i + 1))$. $\square$

**Displacement Sort Complexity in Index2Sort** Next, to analyze the time complexity of sorting $v$ in Index2Sort using the modified Displacement Sort described above, we present the following lemma.

**Lemma A.3.** *In Index2Sort, when the results of approximate rank queries on $u'$ (with at most $\varepsilon$ error) are used as predictions, the expected time complexity of sorting $v$ using the modified Displacement Sort is $\mathcal{O}(n + n\log(\varepsilon + 1))$.*

*Proof of Theorem A.3.* In Index2Sort, the length of the array $v$ to be sorted by the modified Displacement Sort is $n - m$. Let $l = n - m$.

Let $\hat{p} \in \{1, 2, \ldots, m+1\}^l$ be the vector of approximate rank query results on $u'$, and let $p \in \{1, 2, \ldots, m+1\}^l$ be the vector of exact rank query results. Since the approximate rank query has at most $\varepsilon$ error, we have $|\hat{p}_i - p_i| \leq \varepsilon$ for any $i \in \{1, 2, \ldots, m+1\}$.

Let $\eta$ be defined as in Theorem A.2, where $\eta_i = |\{v_j \mid j \in \{1, 2, \ldots, l\} \wedge v_i \leq v_j \wedge \hat{p}_j \leq \hat{p}_i\}|$. The time complexity of the modified Displacement Sort is $\mathcal{O}(l + \sum_{i=1}^{l} \log(\eta_i + 1))$.

Next, we bound $\eta_i$ as follows:

$$\eta_i = |\{v_j \mid j \in \{1, 2, \ldots, l\} \wedge v_i \leq v_j \wedge \hat{p}_j \leq \hat{p}_i\}| \quad (40)$$
$$\leq |\{v_j \mid j \in \{1, 2, \ldots, l\} \wedge v_i \leq v_j \wedge p_j \leq p_i + 2\varepsilon\}| \quad (41)$$
$$\leq |\{v_j \mid j \in \{1, 2, \ldots, l\} \wedge p_i \leq p_j \leq p_i + 2\varepsilon\}| \quad (42)$$
$$\leq \sum_{r=p_i}^{p_i+2\varepsilon} |\{v_j \mid j \in \{1, 2, \ldots, l\} \wedge p_j = r\}|. \quad (43)$$

The first inequality uses the fact that $\hat{p}_j$ differs from $p_j$ by at most $\varepsilon$, while the second uses $v_i \leq v_j \Rightarrow p_i \leq p_j$.

Now, consider $|\{v_j \mid j \in \{1, 2, \ldots, l\} \wedge p_j = r\}|$, i.e., the number of unique elements in $v$ whose exact rank in $u'$ is $r$. Let the indices of elements selected as $u$ in the sorted array $x'$ be $i_1, i_2, \ldots, i_m$, where $1 \leq i_1 < i_2 < \cdots < i_m \leq n$. Additionally, let $i_0 = 0$, $i_{m+1} = n + 1$, $x'_0 = -\infty$, and $x'_{n+1} = \infty$. Then, the value $|\{v_j \mid j \in \{1, 2, \ldots, l\} \wedge p_j = r\}|$ can be bounded as follows:

$$|\{v_j \mid j \in \{1, 2, \ldots, l\} \wedge p_j = r\}| \leq |\{x_j \mid j \in \{1, 2, \ldots, n\} \wedge x'_{i_{r-1}} \leq x_j < x'_{i_r}\}| \quad (44)$$
$$\leq |\{x_j \mid j \in \{1, 2, \ldots, n\} \wedge x'_{i_{r-1}} < x_j < x'_{i_r}\}| + 1 \quad (45)$$
$$\leq i_r - i_{r-1}. \quad (46)$$

Since $\mathbb{E}[i_r - i_{r-1}] = \frac{n}{m+1}$ and $m = \lfloor \alpha n \rfloor$, we have $\mathbb{E}[i_r - i_{r-1}] = \mathcal{O}(1)$. Thus, we have $\mathbb{E}[|\{v_j \mid j \in \{1, 2, \ldots, l\} \wedge p_j = r\}|] = \mathcal{O}(1)$.

From the above, we know $\mathbb{E}[\eta_i] = \sum_{r=p_i}^{p_i + 2\varepsilon} \mathcal{O}(1) = \mathcal{O}(\varepsilon)$. Therefore, the expected time complexity of the modified Displacement Sort is:

$$\mathbb{E}\left[\mathcal{O}\left(l + \sum_{i=1}^{l} \log(\eta_i + 1)\right)\right] = \mathcal{O}(l) + \mathcal{O}\left(\sum_{i=1}^{l} \mathbb{E}[\log(\eta_i + 1)]\right) \tag{47}$$

$$\leq \mathcal{O}(l) + \mathcal{O}\left(\sum_{i=1}^{l} \log(\mathbb{E}[\eta_i] + 1)\right) \tag{48}$$

$$= \mathcal{O}(l) + \mathcal{O}\left(\sum_{i=1}^{l} \log(\varepsilon + 1)\right) \tag{49}$$

$$= \mathcal{O}(l + l\log(\varepsilon + 1)) \tag{50}$$

$$= \mathcal{O}(n + n\log(\varepsilon + 1)). \tag{51}$$

$\square$

**Proof of Theorem 3.4** Using the above lemmas, we now provide the proof of Theorem 3.4 for both cases where modifications (i) and (ii) are applied.

*Proof of Theorem 3.4 (i).* Let $S(n)$ be the time complexity of Index2Sort when modification (i) is applied. In this case, the complexities of the steps ①, ②, ③, ④, and ⑥ remain exactly the same as in Theorem 3.1. For the step ⑤, the time complexity of sorting $v$ using the approximate rank query results as predictions is $\mathcal{O}(n + n\log(\varepsilon + 1))$, as shown in Theorem A.3. Therefore, $S(n)$ can be expressed recursively as follows:

$$S(n) = S(\alpha n) + \mathcal{O}(\alpha n C(\alpha n) + (1 - \alpha)n Q(\alpha n) + n + n\log(\varepsilon + 1)). \tag{52}$$

By applying mathematical induction in the same way as in the proof of Theorem 3.1, we conclude that $S(n) = \mathcal{O}(nC(n) + nQ(n) + n\log(\varepsilon + 1))$. $\square$

*Proof of Theorem 3.4 (ii).* Let $S(n)$ be the time complexity of Index2Sort when modification (ii) is applied. In this case, the complexities of the steps ①, ②, ③, ④, and ⑥ remain exactly the same as in Theorem 3.1.

For the step ⑤, the time complexity of performing the exponential search for each element is $\mathcal{O}((1 - \alpha)n\log(\varepsilon + 1))$, because the difference between the approximate rank query result and the true rank query result is at most $\varepsilon$. Additionally, the total time complexity of sorting each range bucket is $\mathcal{O}(n)$ from Theorem A.1.

Therefore, $S(n)$ can be expressed recursively in the same form as Equation (52). Consequently, by following the same steps as in the proof of Theorem 3.4 (i), we conclude that $S(n) = \mathcal{O}(nC(n) + nQ(n) + n\log(\varepsilon + 1))$. $\square$

A.3    PROOF OF THEOREM 3.5

*Proof of Theorem 3.5.* Let $S(n)$ denote the worst-case time complexity of Index2Sort. Following an approach similar to the proof of Theorem 3.1, the time complexity of each step in the Index2Sort algorithm is as follows:

① The worst-case complexity of splitting the array (including the optional shuffle) is $\mathcal{O}(n)$, as in Theorem 3.1.

② Sorting $u$ recursively takes $S(\alpha n)$ in the worst case.

③ Constructing the index on $u$ has a worst-case complexity of $\mathcal{O}(\alpha n C(\alpha n))$.

④ Answering rank queries for all elements of $v$ using the index has a worst-case complexity of $\mathcal{O}((1 - \alpha)n Q(\alpha n))$.

⑤ Sorting $\boldsymbol{v}$ with bucket sort using the rank query results has a worst-case complexity of $\mathcal{O}(R((1-\alpha)n))$.

⑥ Merging $\boldsymbol{u}'$ and $\boldsymbol{v}'$ has a worst-case complexity of $\mathcal{O}(n)$.

In the step ⑤, one of the worst-case scenarios occurs when $\Omega((1-\alpha)n)$ elements are placed into a single range bucket. In this case, the time complexity of sorting that range bucket is $\Omega(R((1-\alpha)n))$. This represents the worst-case scenario due to the superadditivity of $R(n)$.

Thus, $S(n)$ can be expressed recursively as follows:

$$S(n) = S(\alpha n) + \mathcal{O}(\alpha n C(\alpha n) + (1-\alpha)nQ(\alpha n) + R((1-\alpha)n)). \tag{53}$$

Using the superadditivity of $R(n)$ and following the same mathematical induction approach as in the proof of Theorem 3.1, we conclude that $S(n) = \mathcal{O}(nC(n) + nQ(n) + R(n))$. □

## B  A GENERALIZED FRAMEWORK FOR "ALGORITHMS WITH PREDICTORS"

In this section, we outline an initial method for applying the techniques and theoretical frameworks of algorithms with predictions to problem settings where only the training and inference algorithms of the machine learning model are provided as an opaque box. Specifically, consider a task $T$ where, given a data sequence $\boldsymbol{x}$, the goal is to derive its corresponding ground truth $\boldsymbol{x}'$ ($\boldsymbol{x}$ and $\boldsymbol{x}'$ do not necessarily have the same number of elements). For the sorting problem, $\boldsymbol{x}$ represents the input array, and $\boldsymbol{x}'$ is the sorted version of $\boldsymbol{x}$. In this problem setting, assume the existence of the following algorithms:

- **Predictor Training Algorithm.** For sufficiently large $n$, given a task with $n$ elements and its ground truth, a "predictor" can be trained with a time complexity of $\mathcal{O}(nC(n))$. This predictor satisfies the following properties: given a task with $m$ elements, it can output predictions with time complexity of $\mathcal{O}(mQ(n, m))$, and the "error" of the predictions is at most $\varepsilon$.

- **Algorithm with Predictions.** For sufficiently large $n$ and any $\eta \geq 0$, given a task with $n$ elements and predictions for each element (with a maximum "error" of $\eta$), the task can be completed with time complexity of $\mathcal{O}(P(n, \eta))$.

- **Greedy Algorithm.** For any $n$, given a task with $n$ elements, the ground truth can be obtained in finite time.

Here, the "predictor" does not necessarily need to utilize machine learning; a simpler structure is sufficient. Additionally, the "error" is assumed to be a scalar value defined by an appropriate metric for the specific problem. For example, in the sorting problem, we can define the "error" $\eta := \sum_{i=1}^{n} \log(\eta_i^\Delta + 2)$, where $\eta_i^\Delta$ denote the error between the actual sorted position and the predicted position of the $i$-th element. Under this definition, the time complexity of the Displacement Sort proposed in (Bai & Coester, 2023) is $\mathcal{O}(\eta)$. That is, in this case, $P(n, \eta) = \mathcal{O}(\eta)$.

Here, let the function $C(n)$ be a non-decreasing function of $n$, and let $Q(n, m)$ be a non-decreasing function of both $n$ and $m$. This assumption reflects the natural idea that as the number of data points increases, the computational cost per element for training or inference of the predictor also increases. Note that the computational cost itself does not necessarily need to be monotonic; the assumption of monotonicity applies only to the upper-bound expression.

Similarly, let the function $P(n, \eta)$ be a non-decreasing function of both $n$ and $\eta$. This implies that the time complexity of an algorithm with predictions increases with the number of input elements or with larger errors in the predictions, which is a natural assumption. Again, this monotonicity assumption applies only to the upper-bound expression and does not require the time complexity itself to be strictly monotonic.

Additionally, let $P(n, \eta)$ be a superadditive function with respect to $n$. Specifically, for any $n_1 \geq 0$, $n_2 \geq 0$, and $\eta \geq 0$, we have $P(n_1 + n_2, \eta) \geq P(n_1, \eta) + P(n_2, \eta)$. This assumption reflects the natural notion that the time complexity required to solve a task with $n$ elements is at least as large as the sum of the complexities required to solve two subproblems split from the original task. Again,

---

**Algorithm 2** Algorithm-With-Predictors

1: **Algorithms:**
2:   $\mathcal{A}_c$: Predictor Training Algorithm.
3:   $\mathcal{A}_p$: Algorithm with Predictions.
4:   $\mathcal{A}_g$: Greedy Algorithm.
5:
6: **function** ALGORITHM-WITH-PREDICTORS($\boldsymbol{x}$)
7:     $n \leftarrow |\boldsymbol{x}|$
8:     **if** $n < \tau$ **then**
9:         **return** $\mathcal{A}_g(\boldsymbol{x})$
10:    $\boldsymbol{u} \leftarrow \boldsymbol{x}[1 : \lfloor n/2 \rfloor]$         ▷ ①
11:    $\boldsymbol{u}' \leftarrow$ ALGORITHM-WITH-PREDICTORS($\boldsymbol{u}$)   ▷ ②
12:    $\mathcal{I} \leftarrow \mathcal{A}_c(\boldsymbol{u}, \boldsymbol{u}')$         ▷ ③
13:    $\hat{\boldsymbol{p}} \leftarrow \mathcal{I}.\text{predict}(\boldsymbol{x})$       ▷ ④
14:    **return** $\mathcal{A}_p(\boldsymbol{x}, \hat{\boldsymbol{p}})$      ▷ ⑤

---

superadditivity is assumed for the upper-bound expression, not necessarily for the time complexity itself.

Under these conditions, the following theorem holds:

**Theorem B.1.** *For a task $T$ as defined above, suppose the three algorithms described earlier exist. Then, given data with $n$ elements (without any accompanying predictions or ground truth), there exists an algorithm that can derive the ground truth with a time complexity of $\mathcal{O}(nC(n)+nQ(n,n)+P(n,\varepsilon))$.*

*Proof of Theorem B.1.* Here, the proof is constructive. First, we define the algorithm and then provide proof of its time complexity guarantees.

We define the necessary notation. Let $\mathcal{A}_c$ denote the predictor training algorithm, $\mathcal{A}_p$ denote the algorithm with predictions, and $\mathcal{A}_g$ denote the greedy algorithm. The input data sequence is denoted as $\boldsymbol{x}$, containing $n$ elements.

The algorithm, referred to as *Algorithm-With-Predictors*, is fundamentally similar to Index2Sort. The pseudocode for Algorithm-With-Predictors is presented in Algorithm 2. When the number of elements $n$ in the input data sequence is less than a constant $\tau$, the algorithm uses $\mathcal{A}_g$ to obtain the ground truth. For cases where $n \geq \tau$, the algorithm proceeds as follows:

① Extract half of the data from the input sequence $\boldsymbol{x}$ to create a new sequence $\boldsymbol{u}$.

② Recursively call Algorithm-With-Predictors on $\boldsymbol{u}$ to obtain the ground truth $\boldsymbol{u}'$ for $\boldsymbol{u}$.

③ Call $\mathcal{A}_c$ with $\boldsymbol{u}$ and $\boldsymbol{u}'$ to train a "predictor."

④ Use the trained predictor to make predictions $\hat{\boldsymbol{p}}$ for $\boldsymbol{x}$.

⑤ Call $\mathcal{A}_p$ with $\boldsymbol{x}$ and $\hat{\boldsymbol{p}}$ to obtain the ground truth for $\boldsymbol{x}$.

Now, we give the theoretical guarantees on the time complexity of Algorithm-With-Predictors. Let the time complexity of Algorithm-With-Predictors be $S(n)$. We prove that $S(n) = \mathcal{O}(nC(n) + nQ(n,n) + P(n,\varepsilon))$. The time complexity of each step in the Algorithm-With-Predictors algorithm is as follows:

① Extract half of the data has a complexity of $\mathcal{O}(n)$.

② Recursively calling Algorithm-With-Predictors on $\boldsymbol{u}$ takes $S(n/2)$.

③ Training a predictor with $\boldsymbol{u}$ and $\boldsymbol{u}'$ has a complexity of $\mathcal{O}((n/2)C(n/2))$ (based on the assumptions for $\mathcal{A}_c$).

④ Making predictions $\hat{p}$ for $x$ using the predictor has a complexity of $\mathcal{O}(nQ(n/2, n))$ (based on the assumptions for $\mathcal{A}_c$).

⑤ Deriving the ground truth for $x$ using $x$ and $\hat{p}$ has a complexity of $\mathcal{O}(P(n, \varepsilon))$ (based on the assumptions for $\mathcal{A}_p$).

For the complexity guarantee in ⑤, the fact is used that the "error" in the predictions obtained by ④ is at most $\varepsilon$ from the assumptions for $\mathcal{A}_c$. Thus, $S(n)$ can be expressed recursively as follows:

$$S(n) = S(n/2) + \mathcal{O}((n/2)C(n/2) + nQ(n/2, n) + P(n, \varepsilon)). \tag{54}$$

Using the superadditivity of $P$ and following a similar mathematical induction argument as in the proof of Theorem 3.1, we conclude $S(n) = \mathcal{O}(nC(n) + nQ(n, n) + P(n, \varepsilon))$. □

## C  THEORETICAL GUARANTEE FOR ESPC-INDEX

Here, we provide proof for the following time complexity guarantees of ESPC-index, which were not explicitly mentioned in the original paper (Croquevielle et al., 2025).

**Theorem C.1.** *The ESPC-index, with appropriately adjusted parameters, satisfies $C(n) = 1$ and $Q(n) = \log \log n$ under the assumption that $\boldsymbol{D} \overset{\text{iid}}{\sim} \chi \in \mathfrak{X}_C$ and queries are independently drawn from the same distribution $\chi$. That is, the expected time complexity for construction is $\mathcal{O}(n)$, and the expected time complexity for a single rank query is $\mathcal{O}(\log \log n)$.*

*Proof of Theorem C.1.* The proof follows a similar approach to the proof of Theorem 10 in the original ESPC-index paper (Croquevielle et al., 2025). In Theorem 10, it is shown that for an ESPC-index with parameter $K$ (representing the number of "subintervals" in the ESPC-index), the expected time complexity for construction is $\mathcal{O}(n + K)$, and the expected time complexity for a single rank query is $\mathcal{O}\left(\log \frac{n \log n}{K}\right)$.

In Theorem 10 of (Croquevielle et al., 2025), the time complexities are analyzed for $K = n \log n$. If we instead consider $K = n$, the expected time complexity for construction becomes $\mathcal{O}(n)$, and the expected time complexity for a single rank query becomes $\mathcal{O}(\log \log n)$. □

## D  ADDITIONAL EXPERIMENTAL RESULTS

Here, we present additional experimental results that were not included in the main text due to space constraints. We provide detailed measurements of the number of comparisons in step ⑤ across a wide range of data distributions in Appendix D.1. We then present wall-clock comparisons against both classical and learned sorting algorithms in Appendix D.2.

**Datasets.**  We evaluated our algorithms on both synthetic distributions and a diverse set of real-world datasets.

For artificial data, we used the following four distributions and their shifted versions: uniform distribution on $[0, 1]$, normal distribution with parameters $\mu = 0, \sigma = 1$, exponential distribution with parameter $\lambda = 1$, and log-normal distribution with parameters $\mu = 0, \sigma = 1$. The distribution shift was performed by adding $i/n$ to the $i$-th element, as in the main text.

For real-world data, we used 16 datasets: **Chicago [Start, Tot]**: Taxi trip records reported to the City of Chicago over the last six years, from which we extracted trip start times and total fares (Chicago, 2021). **NYC [Pickup, Dist, Tot]**: New York City yellow taxi trip records, including pickup timestamps, trip distances, and total fare amounts (nyc, 2020). **SOF [Humidity, Pressure, Temperature]**: A time-series of air quality sensor measurements (humidity, pressure, temperature) recorded every minute in Sofia, Bulgaria (Mavrodiev, 2019). **Wiki**: Wikipedia article edit timestamps (Marcus et al., 2020a). **OSM**: Uniformly sampled OpenStreetMap locations, represented as Google S2 CellIds (Marcus et al., 2020a). **Books**: Amazon book sales popularity data (Marcus et al., 2020a). **Face**: An upsampled collection of Facebook user IDs obtained via random walks on the social graph (Marcus et al., 2020b), discarding outliers above the 0.99999 quantile as in (Kristo et al.,

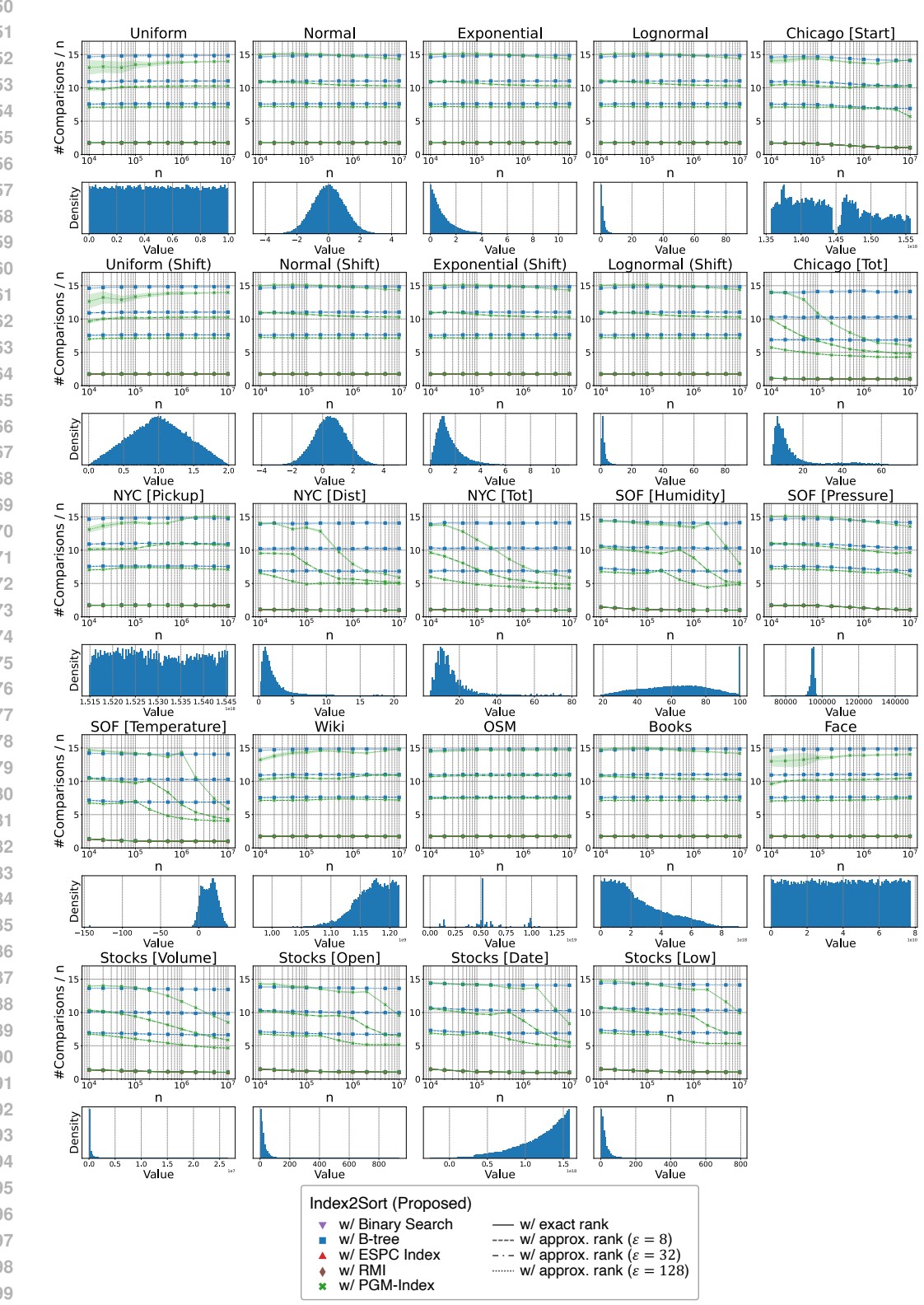

Figure 4: Number of element comparisons required to sort an array of length $n$. Regardless of the distribution, the type of index used, or the precision of rank queries (whether exact or approximate), the number of comparisons required for ⑤ in Index2Sort is observed to be linear with respect to $n$.

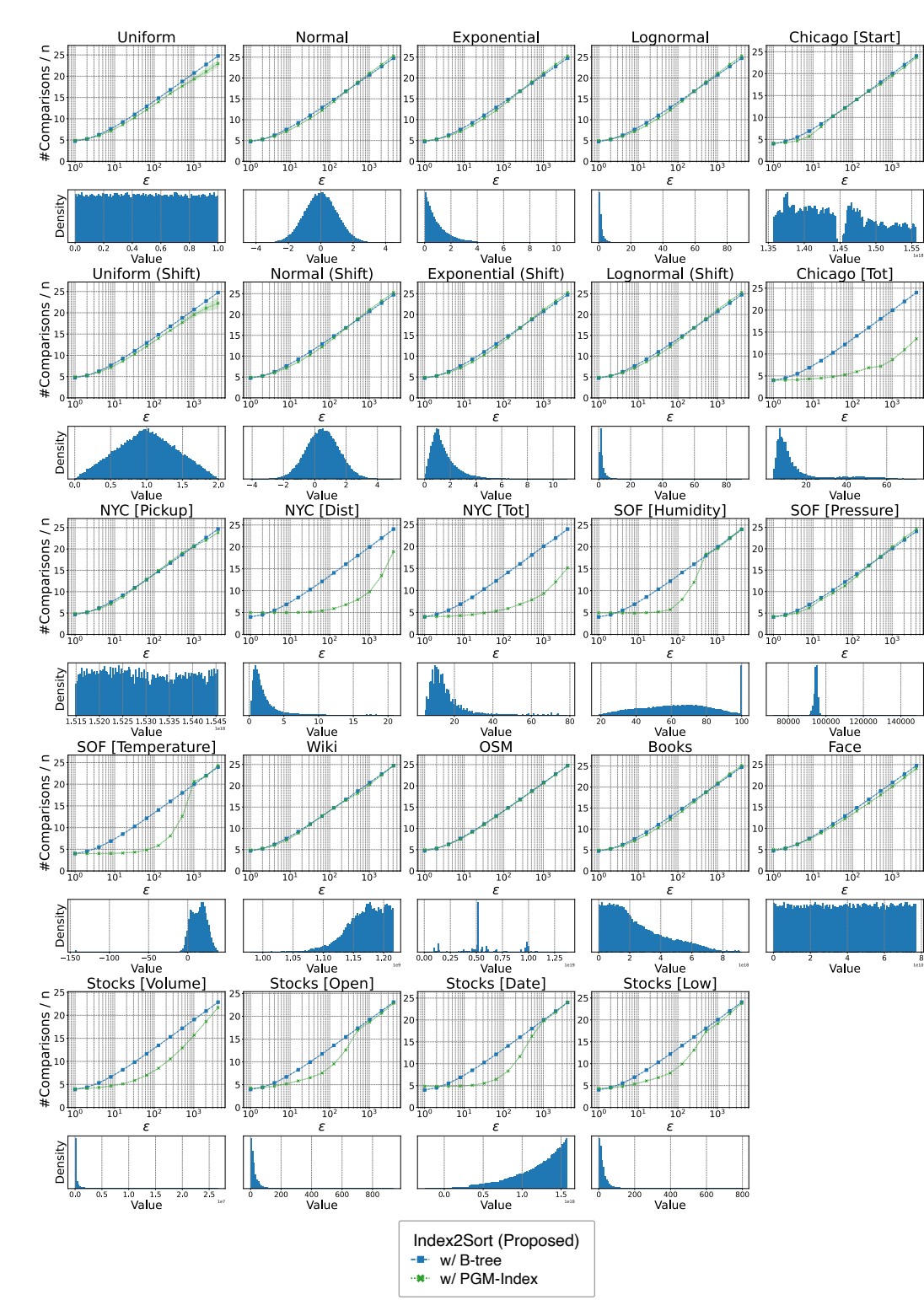

Figure 5: Number of element comparisons in Index2Sort when using an index with a maximum error of $\varepsilon$. Regardless of the distribution or the type of index used, the number of comparisons required for ⑤ in Index2Sort is observed to be proportional to $\log \varepsilon$.

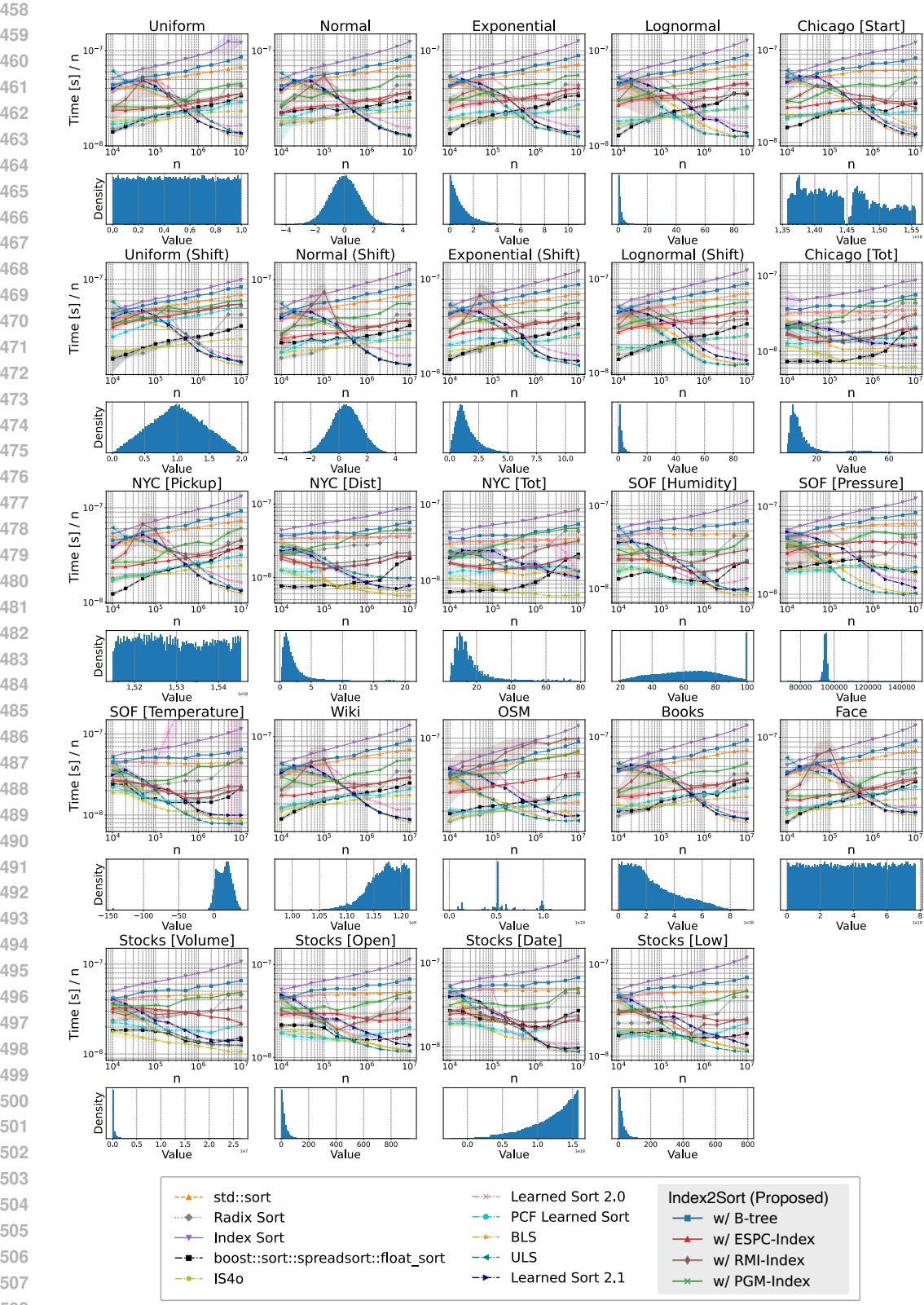

Figure 6: The time consumed to sort an array of length $n$. Index2Sort using the ESPC Index consistently achieves $o(n \log n)$ time complexity. In contrast, Learned Sort 2.0 can degrade to $\mathcal{O}(n^2)$ on real-world data (SOF [Temperature]).

2021; Ferragina & Odorisio, 2025). **Stocks [Volume, Open, Date, Low]**: Historical NASDAQ daily data, including trading volumes, opening and low prices, and dates, retrieved with the `yfinance` Python package (up to April 1, 2020) (Onyshchak, 2020). We generated input arrays by randomly sampling $n$ elements from these datasets. For some datasets (Chicago [Start, Tot], NYC [Dist, Tot], SOF [Humidity, Pressure, Temperature], and Stocks [Volume, Open, Date, Low]), the fraction of unique values is very small (below $3.2\%$), which may affect algorithmic behavior, particularly for methods sensitive to duplicate keys.

**Setup.**    Furthermore, we added additional types of indexes used by Index2Sort. In the main text, we used B-trees, ESPC-index (Croquevielle et al., 2025), and PGM-index (Ferragina & Vinciguerra, 2020), and here we added Binary Search (i.e., performing binary search at query time without constructing an index) and RMI (Kraska et al., 2018). We report the average and standard deviation over 10 runs for each data point in the figures. All experiments were implemented in C++ and conducted on a Linux machine equipped with an Intel® Core™ i9-11900H CPU @ 2.50 GHz and 64 GB of memory. The code was compiled using GCC version 9.4.0 with the `-O3` optimization flag.

### D.1    NUMBER OF COMPARISONS IN STEP ⑤

**Linearity in $n$.**    Figure 4 shows the relationship between the input length $n$ and the number of comparisons performed in step ⑤ of Index2Sort. In this experiment, approximate ranks were obtained using exponential search. For each distribution, we also provide a histogram of the input values. In most cases, the number of comparisons grows almost perfectly linearly with $n$. When using a B-tree as the underlying index, this linear trend is nearly exact. In contrast, with a PGM-index, the number of comparisons sometimes grows at a rate slower than linear. This effect is especially pronounced for datasets with many duplicate values, as the PGM-index resolves duplicates more efficiently than a B-tree and answers approximate rank queries with fewer errors. This improved accuracy reduces the cost of exponential search in step ⑤. Overall, these results indicate that the number of comparisons in step ⑤ is $O(n)$, regardless of the distribution (including shifted and real-world data), the index type, or whether rank queries are exact or approximate.

**Proportionality to $\log \varepsilon$.**    Figure 5 reports the number of comparisons when using a B-tree or a PGM-index under different maximum allowed errors $\varepsilon$. As before, approximate ranks were computed via exponential search, and histograms of the data distributions are shown alongside the results. We observe that, in most cases, the number of comparisons scales proportionally to $\log \varepsilon$. For the B-tree, this proportionality is nearly exact. With a PGM-index, particularly on datasets with many duplicates, the number of comparisons can again be markedly smaller than in the B-tree case. This is because the PGM-index leverages duplicates to improve prediction accuracy, reducing the cost of exponential search in step ⑤. These results demonstrate that the number of comparisons in step ⑤ can be bounded by $O(\log \varepsilon)$, independent of the distribution, index type.

### D.2    WALL-CLOCK COMPARISONS.

We also measured the actual time taken for sorting. As for Index2Sort, we experimented with several index structures: B-tree, ESPC Index (Croquevielle et al., 2025), RMI (Kraska et al., 2018), and PGM-index (Ferragina & Vinciguerra, 2020).

**Baselines.**    We compare our methods with the following baselines:

- **`std::sort`**: The standard sorting routine provided by the C++ Standard Library. It implements IntroSort (Musser, 1997), which has $\mathcal{O}(n \log n)$ worst-case time complexity.

- **Radix Sort**: A non-comparison-based algorithm that processes elements digit by digit. Its running time is $\mathcal{O}(nw)$, where $w$ is the number of digits per element.

- **Index Sort** (Gurram & Gera, 2011): A special case of Index2Sort that uses binary search to determine ranks, without explicitly constructing an index. To the best of our knowledge, our analysis is first to show that its expected computational complexity is $\mathcal{O}(n \log n)$.

- **`boost::sort::spreadsort::float_sort`** (Ross, 2002): Boost C++ implementation of Spreadsort (Ross, 2002).

- **IS⁴o** (Axtmann et al., 2022): A comparison-based Sample Sort variant with super-scalar optimizations and in-place memory usage, representing the state of the art among non-learned sorting algorithms.

- **Learned Sort 2.0** (Kristo et al., 2021): It is one of the early state-of-the-art learned sorting algorithms, though it comes with no formal theoretical complexity guarantees.

- **Balanced Learned Sort (BLS)**, **Unbalanced Learned Sort (ULS)**, and **Learned Sort 2.1** (Ferragina & Odorisio, 2025): They represent the latest state-of-the-art learned sorting algorithms, but they similarly lack formal theoretical guarantees.

- **PCF Learned Sort** (Sato & Matsui, 2024): It has provable expected complexity $\mathcal{O}(n \log \log n)$ under the assumption that the input follows an i.i.d. distribution $\chi \in \mathfrak{X}_{\rho_1, \rho_2}$, and it also guarantees a worst-case complexity of $\mathcal{O}(n \log n)$.

**Index2Sort Implementation Details.** Index2Sort was instantiated with one of the following four index structures: B-tree, ESPC Index (Croquevielle et al., 2025), RMI (Kraska et al., 2018), and PGM-index (Ferragina & Vinciguerra, 2020). We made several implementation improvements (which do not affect the computational guarantees) to make the Index2Sort algorithm faster in practice. First, we set the hyperparameter $\alpha = 1/32$ to reduce the size of $\boldsymbol{u}$. This is because reducing the size of the index makes its construction faster, and also reduces the time required to answer a query. Next, we assume that the index returns an approximate rank ($\varepsilon = 64$). This is because exact rank queries take a relatively long time. Approximate ranks are handled in the following way. First, they are bucketed according to the approximate rank, and then each bucket is sorted with `std::sort`. Next, insertion sort is performed with an upper bound of the number of swaps of $\Theta(n)$, and then, if necessary, sort the array using the modified Displacement Sort. This is because the modified Displacement Sort is relatively slow in practice, and in many cases it is faster to handle the approximate rank error using only insertion sort. The modified Displacement Sort is only performed when the approximate rank contains a very large error. By using this hybrid algorithm, we can achieve both theoretical guarantees and measured performance.

**Results.** Figure 6 shows the relationship between the input length $n$ and the time required to sort the input array. First, we observe that Index2Sort using the ESPC index is generally faster than Index2Sort instantiated with another index. Its running time grows significantly slower than $n \log n$ (e.g., `std::sort`), suggesting an expected complexity of $o(n \log n)$. In particular, our theoretical guarantees for ESPC-based Index2Sort are well supported by the experimental results on artificial data: when the input follows a uniform distribution (included in the class $\mathfrak{X}_{\rho_f}$), the running time is $\mathcal{O}(n)$, and when the input follows normal, exponential, or log-normal distributions (all included in $\mathfrak{X}_C$), the running time is $\mathcal{O}(n \log \log n)$. We also confirmed that other theoretical guarantees hold in practice, including the fact that Index Sort (a special case of Index2Sort) runs in expected $\mathcal{O}(n \log n)$ time regardless of the distribution, and that Index2Sort using either a PGM-Index or a B-tree has expected $\mathcal{O}(n \log n)$ complexity.

Moreover, we find that Index2Sort faithfully inherits the characteristics of the underlying index. Index2Sort with RMI is generally fast, but becomes slower on the OSM dataset. We attribute this to the fact that, on highly skewed datasets such as OSM, the regression error of the RMI model becomes large, leading to expensive post-processing for handling approximate rank queries. Similarly, we observe that Index2Sort instantiated with the PGM-Index is often faster than Index2Sort with the B-tree, even though both have worst-case time complexity $\mathcal{O}(n \log n)$. This shows that the characteristics of the index (namely, that the PGM-Index generally exhibits superior empirical performance compared to the B-tree) are inherited by the sorting algorithm through Index2Sort.

We found that our ESPC-based Index2Sort is on average slower than IS⁴o (a highly optimized comparison-based sorter) as well as recent state-of-the-art learned sorting algorithms such as BLS, ULS, and Learned Sort 2.1. This is expected, as these algorithms have been carefully optimized for cache efficiency and other low-level performance factors, whereas our implementation prioritizes providing rigorous theoretical guarantees. Designing a highly optimized, cache-aware implementation of Index2Sort remains an important direction for future work.

Furthermore, our experiments also reveal the potential risks of algorithms that lack worst-case complexity guarantees (or have very weak ones). For instance, Learned Sort 2.0 exhibited extremely poor performance on the SOF [Temperature] dataset: for $n = 10^5$, while ESPC-based Index2Sort

completed the sort in at most 0.023 seconds, Learned Sort 2.0 took up to **108.9** seconds. This is an empirical manifestation of its $\mathcal{O}(n^2)$ worst-case complexity on real data, underlining the importance of designing algorithms (such as ours) that come with strong worst-case performance guarantees.

PCF Learned Sort also provides an $\mathcal{O}(n \log n)$ worst-case guarantee, ensuring that it runs reliably fast and never deteriorates into pathological slowdowns. In practice, PCF Learned Sort is often faster than Index2Sort instantiated with ESPC-index or RMI. This difference can be attributed to the larger constant factors in Index2Sort's index construction and inference steps, compared with the simple memory accesses and comparisons that dominate the computation inside PCF Learned Sort.

# E    NON-ASYMPTOTIC ANALYSIS OF CONSTANT FACTORS

In this section, we provide a detailed analysis of the constant factors in the running time of Index2Sort. This analysis clarifies how each component of the algorithm contributes to the overall cost and helps illuminate the practical behavior of the method beyond asymptotic notation.

Here, we adopt the following assumptions for the analysis:

- When a sorted array of length $n$ is given, we assume that building an index on it requires $nC(n) + o(nC(n))$ operations. In the earlier asymptotic notation, the constant factors were hidden inside the big-O term. Here, however, the constants are made explicit inside $C(n)$ (e.g., $C(n) = 2n \log n$).

- We assume that performing a rank query on an index built over a sorted array of length $n$ requires $Q(n) + o(Q(n))$ operations. Again, unlike the previous asymptotic definition where constants were absorbed into big-O, the function $Q(n)$ now exposes the constant factors (e.g., $Q(n) = 2 \log n$).

- As the sorting method for range buckets, we assume that insertion sort is used. When the input order is random, the expected number of comparisons of insertion sort is known to be $n^2/4 + o(n^2)$. In our implementation, the randomness of ordering within each range bucket is justified by the initial shuffle.

- For approximate rank queries, we adopt a simple correction method based on exponential search. When the error is at most $\epsilon$, exponential search requires $2 \log_2(\epsilon+1) + o(\log_2(\epsilon+1))$ comparisons.

Now, we provide a non-asymptotic analysis of the running time of each step of Index2Sort. For every step, we explicitly account for the total cost accumulated over all recursive calls occurring in Step ②.

**Step ①**    Although the pseudocode (Algorithm 1) shows that the array is divided and copied into two arrays, an actual implementation does not need to perform this copy. Thus, this step requires only $\mathcal{O}(1)$ time per recursive call. Since the recursion depth is $\mathcal{O}(\log n)$, the total cost of this step is $\mathcal{O}(\log n)$. As we see later, this cost is negligible compared with the other steps and therefore does not affect the constant factors in the overall running time.

**Step ③**    In Index2Sort, an index is built on arrays of lengths $\alpha^i n$ (for $i = 1, 2, 3, \dots$). Therefore, the total cost of index construction in Index2Sort can be upper-bounded as follows:

$$\sum_{i=1}^{\infty} \left( \alpha^i nC(\alpha^i n) + o(\alpha^i nC(\alpha^i n)) \right) \leq \frac{\alpha}{1-\alpha} nC(\alpha n) + o(nC(\alpha n)). \tag{55}$$

**Step ④**    In Index2Sort, for each array of length $\alpha^i n$ ($i = 1, 2, 3, \dots$) on which an index is built, the algorithm performs $\alpha^{i-1}(1-\alpha)n$ rank queries. Thus, the total cost of rank queries in Index2Sort

can be upper-bounded by

$$\sum_{i=1}^{\infty} \left( \alpha^{i-1}(1-\alpha)nQ(\alpha^i n) + o(\alpha^{i-1}(1-\alpha)nQ(\alpha^i n)) \right) \leq nQ(\alpha n) + o(nQ(\alpha n)). \tag{56}$$

**Step ⑤**    We divide the cost of Step 5 into three components: (1) the cost of correcting approximate rank queries using exponential search, (2) the cost of deciding whether each bucket is a point bucket or a range bucket, and (3) the cost of sorting all range buckets.

(1) At recursion depth $i$, the algorithm receives $\alpha^{i-1}(1-\alpha)n$ query results, each of which may contain an error of at most $\varepsilon$. Thus, the total cost of exponential search is

$$\sum_{i=1}^{\infty} \alpha^{i-1}(1-\alpha)n(2\log_2(\epsilon+1) + o(\log_2(\epsilon+1))) = 2n\log_2(\epsilon+1) + o(n\log_2(\epsilon+1)). \tag{57}$$

(2) At recursion depth $i$, the algorithm performs one comparison for each of the $\alpha^{i-1}(1-\alpha)n$ query results. Thus, the total number of comparisons required for this decision step is

$$\sum_{i=1}^{\infty} \alpha^{i-1}(1-\alpha)n = n. \tag{58}$$

(3) We first consider the cost incurred within a single recursion level. Let $A_i$ $(i = 1, 2, \ldots, m+1)$ denote the size of the $i$-th range bucket. From Appendix A, we have

$$\mathbb{E}\left[ \sum_{i=1}^{m+1} A_i^2 \right] \leq \frac{(n-m)(2n-m)}{m+2} = \frac{(1-\alpha)(2-\alpha)}{\alpha}n + o(n). \tag{59}$$

Because the expected cost of insertion sort is $n^2/4 + o(n^2)$, the expected number of comparisons needed to sort all range buckets at this level is $\frac{(1-\alpha)(2-\alpha)}{4\alpha}n + o(n)$. Accumulating this over all recursive calls yields

$$\frac{2-\alpha}{4\alpha}n + o(n). \tag{60}$$

Combining the three components above, the total cost of Step 5 is

$$\left( \frac{3\alpha+2}{4\alpha} + 2\log_2(\epsilon+1) \right) n + o(n + n\log_2(\epsilon+1)). \tag{61}$$

**Step ⑥**    In Step 6, at recursion depth $i$, the algorithm merges the two array by performing $\alpha^{i-1}n + o(\alpha^{i-1}n)$ comparisons. Summing over all recursion depths, the total number of comparisons required for merging is

$$\frac{1}{1-\alpha}n + o(n). \tag{62}$$

Summing the costs of Steps ① through ⑥, the overall number of operations of Index2Sort is given by

$$\frac{\alpha}{1-\alpha}nC(\alpha n) + nQ(\alpha n) + \left( \frac{3\alpha+2}{4\alpha} + \frac{1}{1-\alpha} + 2\log_2(\epsilon+1) \right) n + \text{(lower-order terms)}. \tag{63}$$

## F   THE USE OF LARGE LANGUAGE MODELS (LLMS)

We used large language models to refine the manuscript and help implement small utility scripts, including simple algorithms and plotting code for our experiments.

