# OpenReview forum: "Index2Sort: Sorting Algorithm Using Static Index Data Structure"
_ICLR.cc/2026/Conference — Submitted to ICLR 2026_

### Official Review · Reviewer_vzfr · 2025-10-22

**Soundness:** 3
**Presentation:** 3
**Contribution:** 2
**Rating:** 2
**Confidence:** 4

**Summary:**

This paper proposes the algorithm Index2sort to derive sorting algorithms from various indexing structures, classical or learned indexes, with computational guarantees based on the complexity of the index structures (construction and querying). In particular when using the best known learned index, the result is a sorting algorithm with an expected complexity of O(n). The same strategy works even if the index structures provide approximations for rank queries under certain constraints. Learned indexes and learned sorts approximate the cumulative distribution function of the data using a machine learning algorithm. The expected O(n) sort beats the best known learned sort of complexity (nloglogn).

**Strengths:**

The paper is an interesting work in combinatorial algorithms and provides strong asymptotic results for learned sorting.

**Weaknesses:**

The paper has virtually nothing to do with representation learning or machine learning. The algorithm uses a learned index as a black box and
nothing else uses learning. The results are theoretical with no clear practical advantages to the current best sorting algorithms.

**Questions:**

Why is this work related to representation learning?

Do the errors in handling approximate rank queries accumulate through the recursive algorithm?

---

> ### Author Response · Authors · 2025-12-01
>
> We sincerely appreciate the reviewer's thoughtful comments and constructive feedback on our work. Below, we respond to each comment.
>
> ## Relevance of Index2Sort to the ICLR Community
>
> > The paper has virtually nothing to do with representation learning or machine learning. The algorithm uses a learned index as a black box and nothing else uses learning.
> > Why is this work related to representation learning?
>
> We would like to clarify that a learned index is a model that learns a mapping from keys to their ranks. In other words, it learns a representation that embeds data into a one-dimensional ordered space. The theory of learned indexes has been an active topic in machine learning venues, including ICLR and ICML. Representative works include:
>
> - Why are learned indexes so effective? [Ferragina et al., ICML 2020]
> - Towards Establishing Guaranteed Error for Learned Database Operations [Zeighami & Shahabi, ICLR 2024]
> - Theoretical Analysis of Learned Database Operations under Distribution Shift through Distribution Learnability [Zeighami & Shahabi, ICML 2024]
>
> Index2Sort builds on this line of research by using the learned rank representation as a computational primitive within a sorting framework. We view this contribution as lying at the intersection of learned representations and algorithm design.
>
> We acknowledge that the primary contribution of this work lies in the design and analysis of the algorithm. However, ICLR has a well-established tradition of accepting work that develops algorithms based on predictions supplied by a learned model while treating the model itself as an opaque black box. This “algorithms with predictions” paradigm has been repeatedly represented at the conference. Examples include:
>
> - Learning-Based Frequency Estimation Algorithms [Hsu et al., ICLR 2019]
> - Learning-Augmented Data Stream Algorithms [Jiang et al., ICLR 2020]
> - Partitioned Learned Bloom Filter [Vaidya et al., ICLR 2021]
> - Learning-based Support Estimation in Sublinear Time [Eden et al., ICLR 2021]
> - Learning-Augmented k-means Clustering [Ergun et al., ICLR 2022]
> - Triangle and Four Cycle Counting with Predictions in Graph Streams [Chen et al., ICLR 2022]
> - Improved Learning-augmented Algorithms for k-means and k-medians Clustering [Nguyen et al., ICLR 2023]
> - New Algorithms for the Learning-Augmented k-means Problem [Huang et al., ICLR 2025]
>
> These papers do not introduce new architectures or training methods. Instead, they focus on designing and analyzing algorithms that leverage predictions produced by a learned model, often shallow models such as trees or linear predictors.
>
> Index2Sort follows the same paradigm. As discussed in Section 5, we abstract the predictor (here, the index) as a black-box module and derive rigorous performance guarantees based solely on its construction and query costs. The resulting theoretical improvements align well with the type of contributions that have historically been valued at ICLR.
>
>
> ## Practical Relevance
>
> > The results are theoretical with no clear practical advantages to the current best sorting algorithms.
>
> The aim of this work is not to outperform the fastest existing sorting implementations. Instead, the main contribution is to introduce a new theoretical framework that transfers the computational guarantees of any static index to sorting, without requiring the inspection or modification of the index's internal structure. This reduction establishes a clean and general connection between indexing and sorting, which guarantees that future advances in static learned indexes will immediately translate into improvements in sorting as well. We believe this conceptual contribution is valuable independent of short-term engineering performance.
>
>
> ## Accumulation of Errors
>
> > Do the errors in handling approximate rank queries accumulate through the recursive algorithm?
>
> In our framework, errors do not accumulate across recursive levels. Each recursive step produces a fully sorted subarray. Once a subproblem is sorted, any approximation error introduced by rank queries during that step is eliminated. As a result, every recursive call starts from an error-free state, and no error accumulation occurs throughout the algorithm.

---

### Official Review · Reviewer_zEcV · 2025-10-30

**Soundness:** 2
**Presentation:** 2
**Contribution:** 2
**Rating:** 2
**Confidence:** 5

**Summary:**

The paper introduces Index2Sort, a general framework that derives sorting algorithms from static index data structures by treating them as opaque boxes. The framework automatically inherits index complexity guarantees: if an index has construction time O(nC(n)) and query time O(Q(n)), Index2Sort achieves O(nC(n) + nQ(n)). When instantiated with the ESPC learned index, it achieves O(n) expected time under distributional assumptions—claimed to be strictly better than existing O(n log log n) learned sorts—creating a future-proof paradigm where indexing advances automatically transfer to sorting.

**Strengths:**

**S1-** Novel theoretical framework: The conceptual inversion of using indexes for sorting is elegant and the opaque-box abstraction is intellectually appealing.

**S2-** Rigorous theoretical analysis: The proofs appear technically sound with careful treatment of expected complexity, worst-case bounds, and handling of approximate rank queries.

**S3-** Generality: The framework applies to classical indexes (B-trees) and learned indexes uniformly, with automatic guarantee transfer.

**Weaknesses:**

**W1 - Limited Theoretical Novelty and Overstated Claims**
The O(n) result follows directly from plugging ESPC-index properties (C(n)=Q(n)=1) into the framework, essentially wrapping existing complexity without fundamental innovation. The "strictly tighter" claim is misleading since O(n) holds only under specific assumptions (Xρf or XC) that are not proven weaker than prior work's Xρ1,ρ2. No formal comparison demonstrates that these distributional assumptions are less restrictive. The "opaque box" abstraction is overstated—the algorithm requires a specific interface (construction on sorted arrays, rank queries), not truly opaque.

**W2 - Unresolved Worst-Case Complexity with R(n) Dependence**
The worst-case bound O(nC(n) + nQ(n) + R(n)) includes an explicit R(n) term for sorting range buckets. In standard settings this is O(n log n), but pathologically large range buckets remain unaddressed. Section 6 inadequately discusses when this dependence becomes problematic or provides algorithmic modifications to remove it. Distribution shift analysis assumes δ is known in advance with no practical guidance for determining it. The initial shuffle requirement assumes random permutability, limiting applicability to streaming or ordered data.

**W3 - Poor Practical Performance Despite Theoretical Advances**
Authors acknowledge Index2Sort is "primarily theoretical" and **slower than IS4o and recent learned sorts** (BLS, ULS, Learned Sort 2.1), significantly undermining practical impact. Missing critical analyses include space complexity (index stores sorted u′ requiring O(n) space), cache complexity crucial for real performance, and guidance on when theoretical advantages materialize. No analysis of constant factors that may make O(n log log n) algorithms faster than O(n) for realistic n values.

**W4 - Selective Experimental Presentation and Misleading Comparisons**
Main paper shows only 4 datasets; full results show mixed performance. Some datasets have <3.2% unique values producing atypical behavior. Highlights Learned Sort 2.0's O(n²) failure but compares against older algorithm (2021) without guarantees—fairer comparison would be PCF Learned Sort (2024) with O(n log log n) guarantees and O(n log n) worst-case. No discussion of whether O(n log log n) algorithms have better constant factors or ablation studies on hyperparameters (α, τ, ε).

**W5 - Incomplete Proofs and Incremental Technical Contributions**
Theorem C.1 proof deferred to "similar approach" without full details. Lemma A.1 adapts Frazer & McKellar (1970)—adding point buckets for duplicates is incremental. These incomplete proofs weaken confidence in theoretical claims.

**W6 - Overclaiming and Insufficient Limitations Discussion**
Phrases like "strictly tighter bound," "future-proof paradigm," and "stronger guarantees" overstate conditional results. Results depend on specific distributional assumptions not proven weaker than prior work. Limitations section is brief and doesn't adequately address failure modes, the R(n) dependence problem, or when the "future-proof paradigm" would provide significant practical benefits. Notation inconsistencies (x′ usage, subscript conventions) reduce clarity.

**W7 - Old Leaned Index Methods**  The referenced methods PGM-Index and RMI are outdated, and many advanced learned index methods were not referenced, compared, or used (e.g., DILI, DobLIX).

**Questions:**

**Q1-** On distributional assumptions: Can you provide a formal comparison showing that Xρf or XC are strictly weaker (or at least comparable to) Xρ1,ρ2? What real-world distributions satisfy one but not the other?

**Q2-** On the R(n) term in worst-case: Can you provide tighter analysis or algorithmic modifications to remove the R(n) dependence? You mention this as future work, but it seems fundamental to the approach.

**Q3-** On practical performance: Given that Index2Sort is slower than existing algorithms in practice, what is the target use case? Is this purely theoretical, or do you envision scenarios where this would be deployed?

**Q4-** On constant factors: The O(n) bound hides constant factors. How do these compare to O(n log log n) algorithms for realistic n (say, 10⁶ to 10⁹)?

**Q5-** On the "opaque box" abstraction: Could you clarify what you mean by "opaque"? You still require a specific interface (rank queries on sorted data). How is this different from prior work that uses learned indexes for sorting?

**Q6-** On distribution shift: How would one choose δ in practice without knowing the data distribution in advance?

**Q7-** On the shuffle: Is the initial shuffle truly necessary, or can it be removed with stronger assumptions?

**Q8-** On space complexity: What is the space overhead compared to in-place algorithms like IS4o?

---

> ### Author Response · Authors · 2025-12-01
>
> We appreciate the reviewer's insightful comments on our work. Below, we respond to each comment.
>
> ## W1 - Limited Theoretical Novelty and Overstated Claims
>
> > The O(n) result follows directly from plugging ESPC-index properties (C(n)=Q(n)=1) into the framework, essentially wrapping existing complexity without fundamental innovation.
>
> We respectfully clarify that enabling this wrapping is precisely the contribution of our work.
>
> Without Index2Sort, transferring the guarantees of a learned index to a sorting algorithm is a labor-intensive process. For example, PCF Learned Sort [Sato and Matsui, TMLR 2025] achieves O(n log log n) complexity by reconstructing a sorting algorithm that mimics the internal structure of a learned index with O(n log log n) construction time and O(log log n) query time [Zeighami and Shahabi, ICML 2023].
>
> In contrast, Index2Sort eliminates the need to inspect or redesign the index's internals. The framework enables direct reuse through a minimal interface, which makes such wrapping straightforward.
>
> [Zeighami & Shahabi, ICLM 2023] On Distribution Dependent Sub-Logarithmic Query Time of Learned Indexing
> [Sato & Matsui, TMLR 2025] PCF Learned Sort: a Learning Augmented Sort Algorithm with O(nloglogn) Expected Complexity
>
>
> > The "strictly tighter" claim is misleading since O(n) holds only under specific assumptions (Xρf or XC) that are not proven weaker than prior work's Xρ1,ρ2. No formal comparison demonstrates that these distributional assumptions are less restrictive.
>
> There is a strict hierarchy among the strength of these assumptions: $\mathfrak{X}_C$ is weaker than $\mathfrak{X}{\rho_f}$, and $\mathfrak{X}{\rho_f}$ is weaker than $\mathfrak{X}{\rho_1, \rho_2}$. Formally,
> - For any $\rho_f$, there exists a constant $C$ such that $\mathfrak{X}{\rho_f} \subset \mathfrak{X}_{C}$.
> - For any $\rho_1, \rho_2$, there exists a constant $\rho_f$ such that $\mathfrak{X}{\rho_1, \rho_2} \subset \mathfrak{X}{\rho_f}$.
>
> The proofs are straightforward, and prior work [Croquevielle et al., ICDT 2025] also does not provide explicit proofs of these relationships, which is why they were omitted in the original version. We have now added the corresponding facts and proofs to the main text (Section 2).
>
> [Croquevielle et al., ICDT 2025] Beyond logarithmic bounds: Querying in constant expected time with learned indexes
>
>
> > The "opaque box" abstraction is overstated—the algorithm requires a specific interface (construction on sorted arrays, rank queries), not truly opaque.
>
> Index2Sort requires only the minimal and standard interface for static indexes: construction on a sorted array and rank queries. Our use of the term “opaque” is intended to emphasize that, unlike prior work on learned sorting, Index2Sort does not rely on any analysis of the index’s internal structure (e.g., what arithmetic operations or branching logic it performs). Instead, it relies solely on the primitive operations that are externally exposed. To the best of our knowledge, this is the most opaque way to define static indexes.

---

> ### Author Response · Authors · 2025-12-01
>
> ## W2 - Unresolved Worst-Case Complexity with R(n) Dependence
>
> > The worst-case bound O(nC(n) + nQ(n) + R(n)) includes an explicit R(n) term for sorting range buckets. In standard settings this is O(n log n), but pathologically large range buckets remain unaddressed.
> > Section 6 inadequately discusses when this dependence becomes problematic or provides algorithmic modifications to remove it.
>
> Even in pathological cases, the cost of handling large range buckets is at most $O(n \log n)$, i.e., no worse than standard comparison-based sorting algorithms. Thus, this dependence is not as pessimistic as it may first appear.
>
> Moreover, our algorithmic framework is inherently modular with respect to the procedure used to sort range buckets; any classical/learned sorting algorithm can be plugged in as the subroutine for sorting range buckets. In other words, $R(n)$ can be instantiated with the state-of-the-art worst-case complexity of any sorting algorithm of choice. Consequently, the fact that the worst-case bound of Index2Sort includes an $R(n)$ term should not be viewed so pessimistically.
>
> > Distribution shift analysis assumes δ is known in advance with no practical guidance for determining it.
>
> Index2Sort does not require $\delta$ to be known in advance. The dynamic learned index of [Zeighami & Shahabi, ICML 2024] does not assume prior knowledge of $\delta$. Therefore, when combined with this index, Index2Sort inherits the same adaptivity and does not need $\delta$ in advance.
>
> [Zeighami & Shahabi, ICML 2024] Theoretical Analysis of Learned Database Operations under Distribution Shift through Distribution Learnability
>
>
> > The initial shuffle requirement assumes random permutability, limiting applicability to streaming or ordered data.
>
> In this paper, we focus on the standard setting of sorting a static input array, which is the most widely studied formulation of the problem. Sorting in streaming environments (e.g., online sorting) or algorithms that adapt to the initial degree of sortedness of the input (e.g., adaptive sorting) are interesting directions for future work, but they fall outside the scope of this paper.
>
>
> ## W3 - Poor Practical Performance Despite Theoretical Advances
>
> > Authors acknowledge Index2Sort is "primarily theoretical" and slower than IS4o and recent learned sorts (BLS, ULS, Learned Sort 2.1), significantly undermining practical impact. Missing critical analyses include space complexity (index stores sorted u′ requiring O(n) space), cache complexity crucial for real performance, and guidance on when theoretical advantages materialize.
>
> The aim of this work is not to outperform the fastest existing sorting implementations. Instead, the main contribution is to introduce a new theoretical framework that transfers the computational guarantees of any static index to sorting, without requiring the inspection or modification of the index's internal structure. This reduction establishes a clean and general connection between indexing and sorting, which guarantees that future advances in static learned indexes will immediately translate into improvements in sorting as well. We believe this conceptual contribution is valuable independent of short-term engineering performance.
>
>
> > No analysis of constant factors that may make O(n log log n) algorithms faster than O(n) for realistic n values.
>
> We have added an analysis of the constant factors in Index2Sort. Please refer to our response to Reviewer JXiT for the detailed explanation.

---

> ### Author Response · Authors · 2025-12-01
>
> ## W4 - Selective Experimental Presentation and Misleading Comparisons
>
> > Main paper shows only 4 datasets; full results show mixed performance.
>
> > Some datasets have <3.2% unique values producing atypical behavior.
>
> The four datasets included in the main paper already capture the essential trends. Although the reviewer notes "mixed performance," our key empirical claim remains consistent across all datasets in the full results. The observation that the cost of step 5 in Index2Sort scales as $O(n \log (\epsilon + 1))$ holds across all datasets, which directly supports our Index2Sort’s overall expected-time bound $O(nC(n) + nQ(n) + n \log (\epsilon + 1))$.
> The datasets we used are widely used in prior sorting research. Heavy-duplication scenarios are realistic and practically important. Even on such datasets, the empirical behavior supporting the $O(n \log (\epsilon + 1))$ bound for step 5 remains consistent.
>
> > Highlights Learned Sort 2.0's O(n²) failure but compares against older algorithm (2021) without guarantees—fairer comparison would be PCF Learned Sort (2024) with O(n log log n) guarantees and O(n log n) worst-case.
>
> PCF Learned Sort is included in our list of baselines in Appendix D.2 and is plotted in the experimental results (Figure 6).
>
> Our intention in emphasizing the extreme slowdown of Learned Sort 2.0 was not to present a selective comparison; rather, we aimed to illustrate how the absence of a theoretical worst-case guarantee can lead to severe $O(n^2)$ degradation on real data.
>
> That said, we agree that a clearer discussion comparing Index2Sort with PCF Learned Sort strengthens the presentation. We will revise the experimental section to clarify this point and avoid potential misunderstanding.
>
>
> ## W5 - Incomplete Proofs and Incremental Technical Contributions
>
> > Theorem C.1 proof deferred to "similar approach" without full details.
>
> The proof is complete. The explanation following that sentence provides the full and rigorous argument.
>
>
> ## W6 - Overclaiming and Insufficient Limitations Discussion
>
> > Phrases like "strictly tighter bound," "future-proof paradigm," and "stronger guarantees" overstate conditional results. Results depend on specific distributional assumptions not proven weaker than prior work. Limitations section is brief and doesn't adequately address failure modes, the R(n) dependence problem, or when the "future-proof paradigm" would provide significant practical benefits. Notation inconsistencies (x′ usage, subscript conventions) reduce clarity.
>
> As noted in our response to W1, we now include explicit proofs showing that our distributional assumptions are strictly weaker than those used in prior work.
>
> Regarding $R(n)$, as explained in our response to W2, the dependence on $R(n)$ rarely causes practical problems because the modularity of Index2Sort allows using any state-of-the-art sorting algorithm to sort each range bucket.
>
> ## W7 - Old Leaned Index Methods
>
> > The referenced methods PGM-Index and RMI are outdated, and many advanced learned index methods were not referenced, compared, or used (e.g., DILI, DobLIX).
>
> Our goal is to demonstrate that Index2Sort seamlessly transfers the computational guarantees of a static index to sorting. To support this claim, we use PGM-index, which is one of the state-of-the-art methods with worst-case guarantees, and ESPC-index, which provides state-of-the-art expected-time guarantees. This choice is sufficient for illustrating the generality of the framework. Please also see our response to Reviewer mJ4m for further discussion.
>
> That said, it is entirely reasonable to examine the empirical behavior of Index2Sort under a broader set of indexes. We are currently running additional experiments with DILI and DobLIX, and we will update the submission with these results during the discussion period if they are completed in time.

---

> ### Author Response · Authors · 2025-12-01
>
> ## Q1- On distributional assumptions:
>
> > Can you provide a formal comparison showing that Xρf or XC are strictly weaker (or at least comparable to) Xρ1,ρ2?
>
> As noted in our response to W1, we now include the formal proofs.
>
> > What real-world distributions satisfy one but not the other?
>
> Regarding real-world distributions, we naturally cannot know the exact underlying generating distributions. However, based on the empirical histograms of all datasets used in our experiments, their tails and overall behavior appear consistent with at least the the weakest assumption, $\mathfrak{X}_C$.
>
>
> ## Q2- On the R(n) term in worst-case:
>
> > Can you provide tighter analysis or algorithmic modifications to remove the R(n) dependence? You mention this as future work, but it seems fundamental to the approach.
>
> As explained in our response to W2, the dependence on $R(n)$ is rarely problematic in practice.
>
>
> ## Q3- On practical performance:
>
> > Given that Index2Sort is slower than existing algorithms in practice, what is the target use case? Is this purely theoretical, or do you envision scenarios where this would be deployed?
>
> While Index2Sort can be slower than highly optimized low-level algorithms such as IS4o when sorting primitive numeric types, its main advantage is that it reduces the number of comparisons to $O(n)$ by leveraging the learned distribution.
>
> We envision that Index2Sort can be effective in scenarios where comparisons are expensive, such as large or complex objects and long strings. In these settings, the reduction in comparison cost may outweigh the inference overhead and provide practical gains.
>
>
> ## Q4- On constant factors:
>
> > The O(n) bound hides constant factors. How do these compare to O(n log log n) algorithms for realistic n (say, 10⁶ to 10⁹)?
>
> We have added an analysis of the constant factors in Index2Sort. Please refer to our response to Reviewer JXiT for the detailed explanation.
>
> In the current implementation, the constant factors of Index2Sort tend to be larger than those of PCF Learned Sort (which runs in O(n log log n)). This is because the overhead of index construction and inference is still heavier than simple memory accesses or comparisons.
>
> ## Q5- On the "opaque box" abstraction:
>
> > Could you clarify what you mean by "opaque"? You still require a specific interface (rank queries on sorted data). How is this different from prior work that uses learned indexes for sorting?
>
> Prior works do not directly use a static index for sorting. Instead, they reconstruct new sorting algorithms by manually reorganizing the internal structure of the index and replicating the logic used to obtain its theoretical guarantees. This requires a deep understanding of the index internals.
>
> In contrast, Index2Sort only needs the interface. Once the interface is provided, the sorting algorithm can be built without needing to know or analyze the internal structure or perform a theoretical analysis of the index. In this sense, we state the static index as an opaque box.
>
>
> ## Q6- On distribution shift:
>
> > How would one choose δ in practice without knowing the data distribution in advance?
>
> As explained in our response to W2, $\delta$ does not need to be given in advance.
>
>
> ## Q7- On the shuffle:
>
> > Is the initial shuffle truly necessary, or can it be removed with stronger assumptions?
>
> Actually, under i.i.d. assumptions, the initial shuffle is not required. Without such assumptions, the shuffle is needed to obtain the theoretical guarantees.
>
> ## Q8- On space complexity:
>
> > What is the space overhead compared to in-place algorithms like IS4o?
>
> Learned-sorting methods necessarily require additional space for the model, so they use more space than in-place algorithms such as IS4o.
>
> For Index2Sort, the overhead is as follows:
> - Index memory usage: If the index built on an array of length $n$ uses $O(M(n))$ memory, then Index2Sort requires $O(M(\alpha n))$ to store the index.
> - Other overhead: Additional $O((1 − \alpha)n)$ space is needed to store the rank-query outputs and the buckets.
>
> For example,
> - When Index2Sort uses B-tree, $M(n) = n \log n$, thus, total overhead is $O((\alpha n) \log (\alpha n) + (1-\alpha) n)$
> - When Index2Sort uses ESPC-tree, $M(n) = n$, thus, total overhead is $O((\alpha n) + (1-\alpha) n) = O(n)$

---

### Official Review · Reviewer_JXiT · 2025-11-03

**Soundness:** 3
**Presentation:** 3
**Contribution:** 3
**Rating:** 8
**Confidence:** 4

**Summary:**

The paper designs a theoretical framework for analyzing sorting methods based on the performance of indexing methods. It generalizes various pervious results, and allows for applying other existing results in learned indexing to learned sorting.

**Strengths:**

- The framework itself is interesting, providing a plug-and-play framework for learned sorting that inherits theoretical properties of learned indexing

- The algorithm to perform sorting is also interesting, sorting and building an index on part of the data and applying the index to sort the rest

- There are some neat theoretical insights. Specifically,  that for fast sorting you only need a good static index that answers queries well from the same distribution as the data it was built on. The latter is a nice sorting specific insight that allows application of the results from Croquevielle et al.

**Weaknesses:**

- In general there is a trivial relationship  between sorting and indexing: if a *dynamic* index can be built on n elements in time C_n then we can also sort the data in O(C_n). From that perspective all results from dynamic indexing trivially  apply to sorting. This paper shows that this can be done from *static indexes* as well, and even if the index is only good at answering queries from the same distribution as the data. This is an interesting result, but the intro should specifically emphasize this. Without this context, the answer to the question "can results on indexing transfer to sorting?" is trivially yes, because building a dynamic index is equivalent to sorting  the data.

- As the paper notes, the theoretical constructs with better asymptotic bounds do not necessarily perform better than practical methods with worse asymptotic bounds. Although one factor is optimized implementation (as the paper notes), the constant factors in time complexity also play a big role (difference between nlogn and nloglogn is very small and for many practical datasizes cnloglog n can be larger than nlog n depending on the constant c). It would be good to discuss non-asymptotic bounds and comparison---to know at what data size which method performs better and not asymptotically.

**Questions:**

NA

---

> ### Author Response · Authors · 2025-12-01
>
> We sincerely thank the reviewer for their positive assessment and constructive comments. Below, we respond to each comment.
>
> ## Clarification Regarding Static Indexes
>
> > This is an interesting result, but the intro should specifically emphasize this. Without this context, the answer to the question "can results on indexing transfer to sorting?" is trivially yes, because building a dynamic index is equivalent to sorting the data.
>
> We appreciate this insightful comment and have revised the introduction accordingly.
>
> As the reviewer noted, transferring results from indexing to sorting is indeed trivial for dynamic indexes. Our work, however, focuses on static indexes, where a substantive difficulty arises because a static index requires a sorted array for its construction. In other words, attempting to leverage a static index for sorting necessarily creates a circular dependency.
>
> To clarify this distinction, we have added a new paragraph (now the third paragraph in the Introduction) that explicitly emphasizes that
> - the trivial reduction applies only to dynamic indexes,
> - static indexes present a fundamentally different and nontrivial obstacle, and
> - historically, static learned indexes (e.g., RMI) precede dynamic ones (e.g., ALEX), making static-index-to-sort transfer the meaningful and relevant setting.
>
> We have also refined the wording throughout the paper to clearly distinguish between static and dynamic indexes. These revisions strengthen the motivation and make the scope and contribution of the work more explicit.

---

> ### Author Response · Authors · 2025-12-01
>
> ## Discussion of Constant Factors and Non-asymptotic Bounds
>
> > the constant factors in time complexity also play a big role
>
> > It would be good to discuss non-asymptotic bounds and comparison---to know at what data size which method performs better and not asymptotically.
>
> We appreciate this insightful suggestion. Following the reviewer’s comment, we now provide a non-asymptotic analysis of the number of operations required by each step of Index2Sort. This analysis offers concrete constant factors that complement the asymptotic bounds and help clarify the crossover points between different methods.
>
> Below we summarize the key findings. Here, we present expected operation counts in a non-asymptotic form by explicitly accounting for all operations occurring within the recursive calls (in step 2).
>
> **Split (step 1)**: The cost is $O(1)$, because in the implementation this step does not perform any operations.
>
> **Index construction (step 3)**: To make the constant factors in the index construction cost explicit, we assume that building an index on a sorted array of size $n$ requires $nC(n) + o(nC(n))$ operations. Then, the total cost across all recursive levels is
>
> $$\frac{\alpha}{1-\alpha} nC(\alpha n) + o(nC(\alpha n)).$$
>
> **Rank queries (step 4)**: Similarly to step 3, we assume that a rank query on an index on an array of size $n$ requires $Q(n) + o(Q(n))$ operations. Then, the total cost across all recursive levels is
>
> $$nQ(\alpha n) + o(nQ(\alpha n)).$$
>
> **Bucket sort (step 5)**: We analyze each component:
>
> - **(5a): Handling approximate rank queries**: Using exponential search, the total number of comparisons required to handle approximate rank queries is $2 (1-\alpha) n \log_2 (\epsilon + 1) + o( n \log_2 (\epsilon + 1))$.
> - **(5b): Determining the bucket type (range or point)**: Since this requires one comparison per element, the total number of comparisons is $n + o(n)$.
> - **(5c): Sorting range buckets**: When insertion sort is used, the total expected number of comparisons for sorting all range buckets is $((2 - \alpha) / (4\alpha)) n + o(n)$.
>
> Combining these components, the total number of comparisons in step 5 is
>
> $$\left(\frac{3 \alpha + 2}{4 \alpha} + 2 \log_2 (\epsilon + 1) \right) n + o(n + n  \log_2 (\epsilon+1) ).$$
>
> **Merge (step 6)**: The total number of comparisons is $1/(1 - \alpha) n  +  o(n)$.
>
>
> Putting everything together, the total expected number of operations performed by Index2Sort is
>
> $$\frac{\alpha}{1-\alpha} nC(\alpha n) + nQ(\alpha n) + \left(\frac{3 \alpha + 2}{4 \alpha} + \frac{1}{1 - \alpha} + 2 \log_2 (\epsilon + 1) \right) n  + \textrm{(lower-order terms)}.$$
>
> The number of comparisons in step 5 is obtained experimentally (Figures 2 and 3 in Section 4; Figures 5 and 6 in Appendix D), and the values closely match the coefficients derived above. Below is a subset of the results:
>
> | $n$       | $\epsilon$ | (5a) Analytical | (5a) Measured | (5b) Analytical | (5b) Measured | (5c) Analytical | (5c) Measured |
> |----------:|:-------------:|----------------:|--------------:|----------------:|--------------:|----------------:|--------------:|
> | 1e5       | 0             | 0               | 0.00e0        | 1.00e5          | 9.99e4        | 7.50e4          | 7.48e4        |
> | 1e5       | 8             | 6.34e5          | 5.87e5        | 1.00e5          | 9.99e4        | 7.50e4          | 7.51e4        |
> | 1e5       | 128           | 1.40e6          | 1.31e6        | 1.00e5          | 9.99e4        | 7.50e4          | 7.51e4        |
> | 1e7       | 0             | 0               | 0.00e0        | 1.00e7          | 1.00e7        | 7.50e6          | 7.51e6        |
> | 1e7       | 8             | 6.34e7          | 5.87e7        | 1.00e7          | 1.00e7        | 7.50e6          | 7.51e6        |
> | 1e7       | 128           | 1.40e8          | 1.31e8        | 1.00e7          | 1.00e7        | 7.50e6          | 7.51e6        |
>
>
> We appreciate the reviewer’s suggestion regarding constant factors. The complete non-asymptotic analysis has been added as Appendix E in the revised version.

---

### Official Review · Reviewer_U1Pt · 2025-11-04

**Soundness:** 3
**Presentation:** 2
**Contribution:** 2
**Rating:** 4
**Confidence:** 4

**Summary:**

This paper studies the sorting problem with static indexes. Their algorithm, Index2Sort, requires a static index built on a sorted array A and supports rank queries on A.

Let the construction time and the per-query time of the static index be $O(n C(n))$ and $Q(n)$, respectively. Index2Sort can sort an array of n elements in $O(n C(n) + n Q(n))$ time. If the index answers approximate rank queries with error $\varepsilon$, the running time becomes $O(n C(n) + n Q(n) + n \log(\varepsilon+1))$. Combining these bounds with an index for which $C(n)=O(1)$ and $Q(n)=O(1)$, they obtain a linear-time sorting algorithm under certain input assumptions.

**Strengths:**

1. The work provides theoretical time bounds for distributions with shifts, approximate rank queries, and worst-case inputs. These results show that Index2Sort is not restricted to ideal cases and is applicable in various practical scenarios.

2. The algorithm is simple and efficient; advances in learned indexes may further improve its performance.

**Weaknesses:**

1. The Index2Sort algorithm and its analysis resemble existing CDF-based learned sorts. A notable difference is that Index2Sort needs a recursive phase to grow a sufficiently large sorted sample for bucketing. Consequently, the contribution is more like an application of state-of-the-art learned indexes than a technical novelty.

2. Although the experiments demonstrate potential—performance is comparable to several existing algorithms—the implementation is not fully optimized for cache efficiency or other low-level factors, and it remains slower than the state-of-the-art sorting implementations.

**Questions:**

The experimental results for Index2Sort with the RMI-index appear incomplete on some datasets (e.g., Stocks [Volume]).

Nearly half of several figures are almost empty, while the remaining half contain 14 lines, making them hard to recognize.

---

> ### Author Response · Authors · 2025-12-01
>
> We sincerely thank the reviewer for the thoughtful assessment and constructive feedback. We address each point in detail below.
>
> ## Similarity to Existing CDF-based Learned Sorts
>
> > The Index2Sort algorithm and its analysis resemble existing CDF-based learned sorts. A notable difference is that Index2Sort needs a recursive phase to grow a sufficiently large sorted sample for bucketing. Consequently, the contribution is more like an application of state-of-the-art learned indexes than a technical novelty.
>
> We respectfully clarify that, although Index2Sort may appear similar to existing CDF-based learned sorts at a superficial level, its core design principle is fundamentally different. The key novelty lies in the fact that a static index algorithm is used directly inside a sorting algorithm without any modification to its internal structure. Index2Sort treats the static index as an opaque module and derives sorting algorithms with state-of-the-art computational guarantees.
> This design enables “an application of state-of-the-art learned indexes” to sorting. Without our framework, such an application would require substantial manual effort. For example, PCF Learned Sort [Sato and Matsui, TMLR 2025] achieves O(n log log n) complexity by reconstructing a sorting algorithm that mimics the internal structure of a learned index with O(n log log n) construction time and O(log log n) query time [Zeighami and Shahabi, ICML 2023]. In contrast, Index2Sort eliminates the need to inspect or redesign the index's internals. The framework enables direct reuse through a minimal interface, which makes such application straightforward.
>
> ## Performance Compared to State-of-the-Art Implementations
>
> > Although the experiments demonstrate potential—performance is comparable to several existing algorithms—the implementation is not fully optimized for cache efficiency or other low-level factors, and it remains slower than the state-of-the-art sorting implementations.
>
> We agree that combining theoretical guarantees with fully optimized, cache-efficient implementations is important. However, the primary contribution of this work is not to surpass current state-of-the-art engineering results. Instead, the goal is to present a general framework that converts a static index into a sorting algorithm, and to obtain sorting algorithms with state-of-the-art theoretical guarantees.
>
>
> ## Questions Regarding Plot Rendering Issues
>
> > The experimental results for Index2Sort with the RMI-index appear incomplete on some datasets (e.g., Stocks [Volume]).
>
> We appreciate the reviewer’s attention to this issue. After further debugging, we identified and corrected the problem affecting the RMI-index experiments. We have now added the missing results, and the updated curves align naturally with the existing results. These corrections do not affect any of the findings reported in the paper.
>
> > Nearly half of several figures are almost empty, while the remaining half contain 14 lines, making them hard to recognize.
>
> We revised the figures by using a logarithmic scale on the y-axis, which makes the curves easier to recognize.

---

### Official Review · Reviewer_i38V · 2025-11-05

**Soundness:** 4
**Presentation:** 4
**Contribution:** 2
**Rating:** 4
**Confidence:** 3

**Summary:**

The paper proposes Index2Sort, a general framework that derives sorting algorithms from static indexes by treating the index as an opaque box supporting only two operations: index construction and rank queries. The core idea is that any index capable of rank queries can be reused for sorting, and its computational guarantees automatically transfer to the sorting process. Specifically, if the index can be constructed in time O(nC(n)) and each rank query can be answered in time O(Q(n)), the resulting sorting algorithm runs in O(nC(n) + nQ(n)) time. Using a state-of-the-art learned index under certain distributional assumptions, the overall complexity can be as fast as O(n). The authors perform experiments to analyze the running time of their framework on both synthetic and real data, including cases with approximate query-answer oracles.

**Strengths:**

1. The proposed algorithm and framework are general and can be applied to many existing learned indexes, as well as future ones.
2. The authors conduct experiments on both real and synthetic datasets, and the results align well with the theoretical complexity analysis.

**Weaknesses:**

1. The algorithm and its analysis are relatively simple and conceptually close to the standard divide-and-conquer framework, which makes the novelty somewhat limited.
2. The theoretical guarantees for learned indexes depend heavily on distributional assumptions, which may limit their applicability in practice.

**Questions:**

n/a

---

> ### Author Response · Authors · 2025-12-01
>
> We sincerely thank the reviewer for the thoughtful assessment and constructive feedback. We address each point in detail below.
>
> ## Simplicity and Novelty of the Proposal and Analysis
>
> > The algorithm and its analysis are relatively simple and conceptually close to the standard divide-and-conquer framework, which makes the novelty somewhat limited.
>
> The core structure of Index2Sort is fundamentally different from a standard divide-and-conquer framework (For this reason, the term divide-and-conquer is intentionally avoided in the paper).
> - Classical sorting algorithms such as Merge Sort and Quick Sort, as well as existing learned sorting methods [Kristo et al., SIGMOD 2020; Zeighami & Shahabi, ICML 2024; Sato & Matsui, TMLR 2025], process recursively generated subproblems in a symmetric manner: each subproblem is handled using the same recursive procedure.
> - In contrast, Index2Sort introduces an asymmetric structure. After splitting the array into two parts, one part is recursively sorted while the other is processed using a bucket sort guided by rank queries. This asymmetric structure enables Index2Sort to transfer the computational guarantees of the underlying index directly to the sorting algorithm.
>
> Although the algorithm is indeed simple, the simplicity is a strength rather than a weakness. It allows Index2Sort to achieve state-of-the-art expected complexity under strictly weaker distributional assumptions (for example, $O(n)$ under $\chi \in \mathfrak{X}_{\rho_f}$). We believe that achieving such guarantees with a clean and conceptually transparent framework represents meaningful novelty.
>
> [Kristo et al., SIGMOD 2020] The Case for a Learned Sorting Algorithm
> [Zeighami & Shahabi, ICML 2024] Theoretical Analysis of Learned Database Operations under Distribution Shift through Distribution Learnability
> [Sato & Matsui, TMLR 2025] PCF Learned Sort: a Learning Augmented Sort Algorithm with O(nloglogn) Expected Complexity
>
>
> ## Dependence on Distributional Assumptions
>
> > The theoretical guarantees for learned indexes depend heavily on distributional assumptions, which may limit their applicability in practice.
>
> Recent theoretical progress on learned indexes has been rapid, and strong guarantees are now available under much weaker assumptions than before. Index2Sort is designed to take advantage of this progress: any performance guarantee proved for a learned index is automatically transferred to sorting, without the need for additional algorithmic design. As these guarantees improve, the guarantee of Index2Sort improves accordingly.
>
> Even with the guarantees currently available in the literature, the assumptions are already practical. For example, when using the ESPC index, Index2Sort achieves an expected running time of $O(n \log \log n)$ under the assumption that the input distribution is sub-exponential ($\chi \in \mathfrak{X}_{C}$). This assumption is much weaker than those required in previous learned sorting algorithms, and it is satisfied by many distributions encountered in real applications.
>
> Finally, the framework is not limited to distribution-dependent guarantees. As shown in Theorem 3.1, the worst-case guarantees of indexes that require no distributional assumptions (e.g., B-trees and PGM-index) are transferred directly to sorting. Therefore, regardless of whether the index is classical or learned, and regardless of whether its guarantee depends on distributional assumptions, the framework consistently produces a sorting algorithm with the appropriate performance guarantee.

---

### Official Review · Reviewer_mJ4m · 2025-11-09

**Soundness:** 2
**Presentation:** 2
**Contribution:** 1
**Rating:** 2
**Confidence:** 4

**Summary:**

The paper presents a sorting algorithm that extends the bucket sort. The core idea is that, given an array, it is first randomly permuted, followed by a portion of it being sorted, then a learned index is created using the sorted subset, and finally, the unsorted portion is bucket sorted using the learned index. Finally, merging both sorted parts. In essence, instead of performing a bucket sort throughout, initially, a learned index is created over a subset to utilize it for expected linear-time sorting of the unsorted part.

**Strengths:**

This is a simple yet interesting idea to employ the learned indices in the hierarchy of the bucket sort.

Additionally, it is attractive to present a framework that can utilize any learned index for such a sorting algorithm.

**Weaknesses:**

This work, though an application of shallow machine learning, does not offer any new insight. Even from an algorithm design perspective, the contributions are only marginal. For the core ML community, such as ICLR, the submission is of little interest.

For someone from the core DBMS community, a natural question to ask would be, if you need only a static index, what prevents you from using HistTree [1], a non-learned index in place of a B-tree, and then critically analyze the performance. Even for a learned index, there are better candidates than the PGM-index, such as LIPP/ALEX (yes, ALEX offers dynamic updates; still, it outperforms the PGM-index for static updates), see [2]. Then, why not use them in your experiments?

The claimed algorithmic innovations in the submission may be of greater interest to communities such as ESA and ICALP. For innovations from a DBMS perspective, ICDT, CIDR, and similar conferences may find more alignment.

[1] Crotty, Andrew. "Hist-Tree: Those Who Ignore It Are Doomed to Learn." CIDR. 2021.

[2] Sun, Zhaoyan, Xuanhe Zhou, and Guoliang Li. "Learned index: A comprehensive experimental evaluation." Proceedings of the VLDB Endowment 16.8 (2023): 1992-2004.

**Questions:**

See the weaknesses. Address those points.

---

> ### Author Response · Authors · 2025-12-01
>
> We sincerely thank the reviewer for their thorough review and constructive feedback on our work. Below, we respond to each comment.
>
> ## Relevance of Index2Sort to the ICLR Community
>
> > This work, though an application of shallow machine learning, does not offer any new insight.
>
> > For the core ML community, such as ICLR, the submission is of little interest.
>
> > The claimed algorithmic innovations in the submission may be of greater interest to communities such as ESA and ICALP. For innovations from a DBMS perspective, ICDT, CIDR, and similar conferences may find more alignment.
>
> We respectfully clarify that algorithmic research that treats ML models, including those that are not necessarily deep, as opaque predictors is well within the scope of ICLR. This line of work has been a sustained theme at the conference, where numerous accepted papers use ML predictors as black-box oracles and develop algorithmic or theoretical contributions on top of them. Notable examples include:
>
> - Learning-Based Frequency Estimation Algorithms [Hsu et al., ICLR 2019]
> - Learning-Augmented Data Stream Algorithms [Jiang et al., ICLR 2020]
> - Partitioned Learned Bloom Filter [Vaidya et al., ICLR 2021]
> - Learning-based Support Estimation in Sublinear Time [Eden et al., ICLR 2021]
> - Learning-Augmented k-means Clustering [Ergun et al., ICLR 2022]
> - Triangle and Four Cycle Counting with Predictions in Graph Streams [Chen et al., ICLR 2022]
> - Improved Learning-augmented Algorithms for k-means and k-medians Clustering [Nguyen et al., ICLR 2023]
> - New Algorithms for the Learning-Augmented k-means Problem [Huang et al., ICLR 2025]
>
> These works do not aim to introduce new model architectures or training methods. Instead, they focus on designing and analyzing algorithms that leverage the predictions provided by an ML model without relying on the ML model’s internal structure. Notably, the models used in the experiments of [Vaidya et al., 2021], [Nguyen et al., 2023], and [Huang et al., 2025] are not deep learning models, but rather classical predictors such as decision trees, linear models, and shallow neural networks. The historical record at ICLR clearly shows that shallow or classical predictors can support substantial algorithmic and theoretical contributions.
>
> The proposed Index2Sort framework fits well within this established line of research. As discussed in Section 5, it follows the algorithm-with-predictions paradigm by abstracting the predictor (in our case, the index) as an opaque module and deriving rigorous performance guarantees. The resulting theoretical improvements align well with the type of contributions repeatedly valued by the ICLR community.
>
>
> ## Choice of Index Structures in the Experiments
>
> > what prevents you from using HistTree [1]
>
> > there are better candidates than the PGM-index, such as LIPP/ALEX
>
> We selected index structures based on the central goal of this work: validating the primary claim of Index2Sort, namely that the computational guarantees of a static index can be directly transferred to sorting. For this purpose, it was important to include representative indexes that are widely used and provide explicit theoretical guarantees on construction and query costs. The original submission therefore included the following:
> - B-tree, the most classical and widely used static index with worst-case complexity guarantees.
> - RMI, the first learned index and still a common baseline.
> - PGM-index, the learned index which provides state-of-the-art worst-case complexity guarantees.
> - ESPC index, the learned index which provides state-of-the-art expected-time complexity guarantees.
>
> The reviewers’ suggestions (HistTree, ALEX, LIPP) are indeed strong practical candidates. However, they are not as well aligned with the core purpose of validating theoretical guarantee transfer:
> - HistTree does not provide explicit complexity bounds for construction or queries.
> - ALEX does not give clear worst-case or expected-time bounds in terms of the input size (Its analysis relies on a parameter $p$, defined in terms of the key distribution, and $p$ can be extremely large).
> - LIPP provides $O(n \log n)$ construction time and $O(n \log n)$ query time, identical in asymptotic order to the static PGM-index already included.
>
> That said, it is entirely reasonable to examine the empirical behavior of Index2Sort under a broader set of indexes. We are currently running additional experiments with HistTree, LIPP, and ALEX, and we will update the submission with these results during the discussion period if they are completed in time.

---

### Meta-Review · Area_Chair_synS · 2025-12-23

**Summary:**

The paper introduces Index2Sort, a framework that transforms any static index data structure into a sorting algorithm. Instantiated with state-of-the-art learned indexes (ESPC), it achieves expected linear time $O(n)$ sorting, improving prior learned sorting bounds of $O(n \log \log n)$.

Strengths:
- The reduction bridges a gap between static indexing and sorting.
- The framework treats the index as a black box, adapting to future advances in indexing automatically.

Weaknesses:
- The recursive overhead and index construction costs result in higher constant factors
- The novelty of the paper should be clarified more explicitly as many reviewers were concern about the incremental nature of the work.

Overall, the paper presents some interesting ideas but it is not ready for publication.

**Reviewer Concerns:**

I think the main critique raised by the reviewers is novelty and it is only partially addressed by the rebuttal.

**Reviewer Scores:**

I think the reviewers would not have change their scores with a longer discussion. In particular, many reviewers seem to agree that the novelty of the paper is limited and the rebuttal is not too convincing in this regard.

---

### Decision · Program_Chairs · 2026-01-26

Reject